theoretical biology, microbiology

CRISPR-Cas, adaptive immunity, bacteriophage, host–virus interactions

**Author for correspondence:**
JL Weissman
e-mail: jakeweis@usc.edu

# Immune lag is a major cost of prokaryotic adaptive immunity during viral outbreaks

JL Weissman[1], Ellinor O. Alseth[2], Sean Meaden[2], Edze R. Westra[2] and Jed A. Fuhrman[1]

[1]Department of Biological Sciences—Marine and Environmental Biology, University of Southern California, Los Angeles, CA, USA
[2]Environment and Sustainability Institute, Biosciences, University of Exeter, Penryn Campus, Penryn, UK

JLW, 0000-0002-4237-4807; ERW, 0000-0003-4396-0354

Clustered regularly interspaced short palindromic repeat (CRISPR)-Cas adaptive immune systems enable bacteria and archaea to efficiently respond to viral pathogens by creating a genomic record of previous encounters. These systems are broadly distributed across prokaryotic taxa, yet are surprisingly absent in a majority of organisms, suggesting that the benefits of adaptive immunity frequently do not outweigh the costs. Here, combining experiments and models, we show that a delayed immune response which allows viruses to transiently redirect cellular resources to reproduction, which we call 'immune lag', is extremely costly during viral outbreaks, even to completely immune hosts. Critically, the costs of lag are only revealed by examining the early, transient dynamics of a host–virus system occurring immediately after viral challenge. Lag is a basic parameter of microbial defence, relevant to all intracellular, post-infection antiviral defence systems, that has to-date been largely ignored by theoretical and experimental treatments of host-phage systems.

## 1. Introduction

Clustered regularly interspaced short palindromic repeat (CRISPR)-Cas immune systems are the only known form of adaptive immunity found in prokaryotic organisms [1,2]. These antiviral defence systems enable bacteria and archaea to incorporate short stretches of viral genetic material into specific loci on the host genome (the CRISPR 'array') as individual immune memories, called 'spacers' [2]. Spacers are later transcribed and processed into CRISPR-RNA (crRNA) sequences that guide CRISPR-associated (Cas) proteins to cleave viral nucleic acids [3–5]. Thus, a genomic record of past infections is used to prevent future infection (see [6] for a recent review of the mechanisms of CRISPR-Cas immunity).

CRISPR-Cas systems are widely but sparsely distributed across the tree of life [7–9]. Their broad distribution among taxa is probably attributable to the fact that these systems are highly effective at clearing viral infections (e.g. [2]), extremely adaptable in a constantly shifting co-evolutionary arms race [10,11], and, similarly to other defence systems [12], frequently horizontally transferred [12–15] (for reviews of various aspects of CRISPR biology, see [9,16–18]). However, a majority of prokaroytes lack CRISPR-Cas immune systems [19], even as CRISPR-Cas can usually be found in a closely related relative. To solve this apparent paradox various authors have proposed a number of costs and limitations of CRISPR-Cas immunity that may drive selection against this system in favour of alternative defence strategies (of which there are many [20]). These 'cons of CRISPR' potentially include autoimmunity [21], the inhibition of beneficial horizontal gene transfer [22,23], the inhibition of other cellular processes by the *cas* genes (specifically DNA repair; [24,25]), incompatibility with lysogenic phage [26], and the possibility that CRISPR-Cas may be unable to keep up with extremely diverse pathogenic environments [27,28]. Nevertheless, experiments show that CRISPR-Cas systems can be essentially cost-free in phage-free culture conditions [29–31].

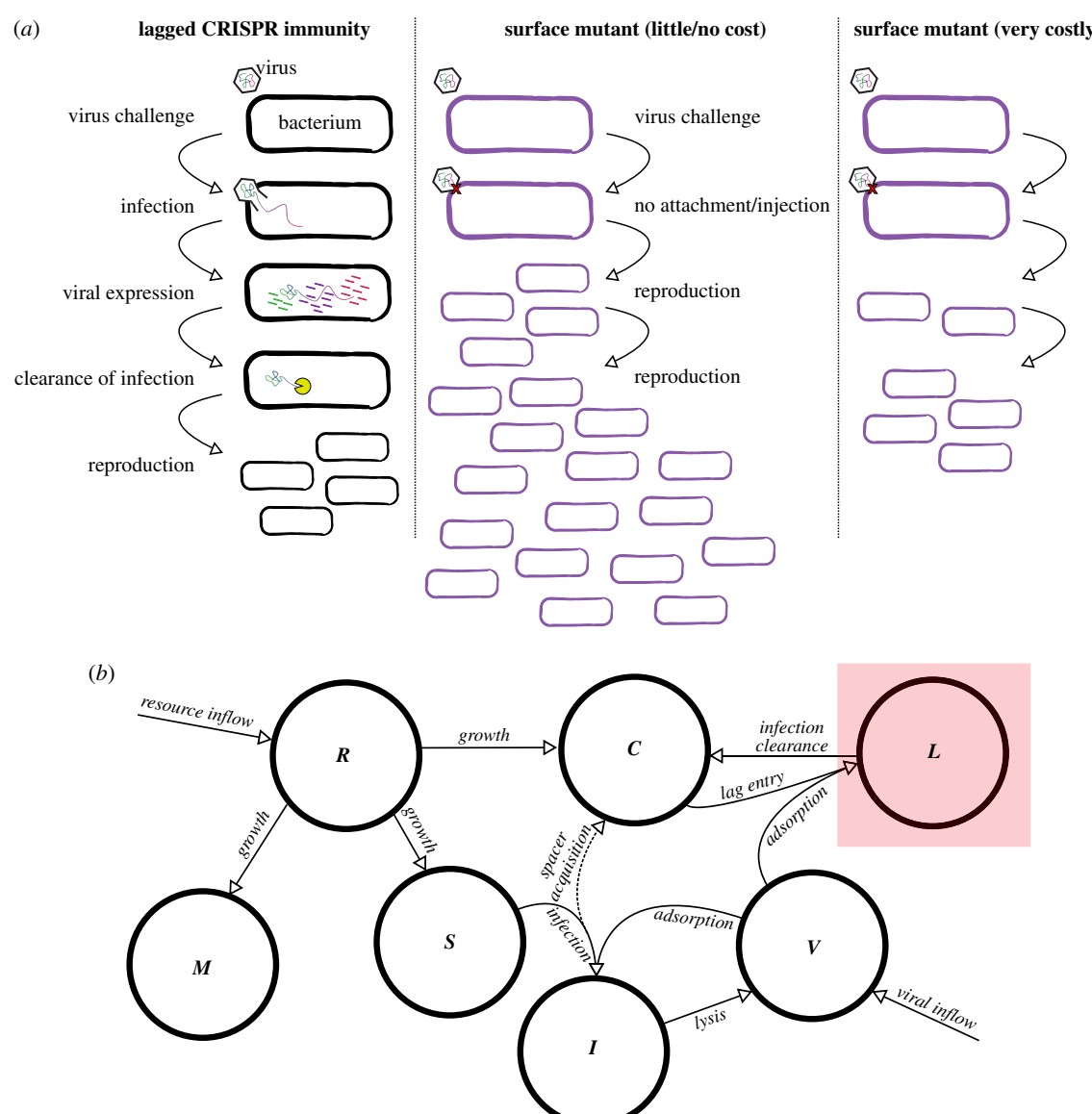

**Figure 1.** Considering lagged CRISPR immunity. (*a*) During competition, immune lag can lead infected cells to face significant delays in reproduction while cellular resources are temporarily diverted to viral production before the infection can be cleared via cleavage of viral nucleic acids (left). By contrast, a surface mutant (SM) strategy does not allow viral genetic material to enter the cell in the first place, preventing any lag (middle). The balance between the cost of lag and any growth cost associated with the SM strategy will determine whether CRISPR immunity is favoured evolutionarily (right). (*b*) Schematic of model of host–virus dynamics with lagged CRISPR immunity. Lagged cell compartment highlighted for emphasis (compare to base model without lag in the electronic supplementary material, figure S1). The dashed line for spacer acquisition signifies that these immunization events are rare. (Online version in colour.)

In contrast to results indicating that CRISPR-Cas defence generates little-to-no constitutive costs for the host in the absence of phage—at least in laboratory-reared *Pseudomonas aeruginosa*—a severe inducible cost of CRISPR-Cas immunity upon phage infection has been observed [29]. The source of the inducible cost of CRISPR-Cas immunity was, until recently, mysterious. Importantly, while CRISPR-immune cells were observed to have reduced fitness when exposed to phage in competition experiments, subsequent efficiency of plating experiments showed that CRISPR-immune cells did not experience a detectable level of phage-induced mortality [29], indicating that phage inhibit the growth of immune cells but either do not kill them or do so very rarely at levels that cannot explain CRISPR's inducible cost. Chabas *et al.* [32] suggested that the inducible cost is the result of transient expression of phage genes in the cell before CRISPR-Cas is able to clear an infection (figure 1). Recently, Meaden *et al.* [31] have provided evidence confirming this lag hypothesis in experiments with *P. aeruginosa* strain PA14 and its phage DMS3*vir*, demonstrating that phage gene expression

is responsible for a reduction in the fitness of CRISPR-immune host. Specifically, Meaden *et al.* [31] showed that a phage protease was transiently expressed in CRISPR-immune cells before infection could be cleared, and that expression of this gene was detrimental to host fitness. When a virus infects a cell, viral genes will be expressed, often at very high levels. At the same time, the host cell's expression patterns may be 'reprogrammed' by the infecting virus (creating a 'virocell'; [33–35]). Intracellular DNA- or RNA-degrading defences may take some time to find and degrade invading genetic material in the cell, and during that time transcription in an infected cell may be transiently altered [34,35], potentially halting host growth and re-purposing cellular resources. This phenomenon, which we call 'immune lag', was observed by Meaden *et al.* [31] in CRISPR-immune cells, so that even when cells are able to effectively clear infections and prevent lysis they still pay a heavy growth cost associated with infection (figure 1*a*). Could immune lag be a major cost of adaptive immunity, leading the host to sometimes favour alternative immune strategies?

Upon closer examination, the impact of immune lag on natural systems is less clear. Experiments that demonstrate the inducible cost of CRISPR-Cas immunity require the host to be exposed to extraordinarily high viral titres (at least $10^8$–$10^{10}$ PFU ml$^{-1}$ in our own experiments, described below) to see any effect. For lag to have any population-level impact on an immune host population, a substantial portion of the immune population must be exposed to phages. Thus, in the case of an already-immunized host population, lag is probably irrelevant because viruses have no way to reach sufficiently high titres to suppress the immune host.

However, in natural systems, host populations are rarely completely immune to their viral pathogens. CRISPR spacers can be lost [15,22,36–39], and viral escape mutants with point mutations in protospacer regions frequently emerge [40,41], both leading CRISPR-Cas to be a somewhat transient form of immunity [31]. In natural communities, entirely new species of virus, to which the host lacks pre-existing immunity, may migrate into the system via dispersal [42]. Thus, to fully characterize the role of immune lag in natural systems, we must assess its impact on non-equilibrium systems with viral coevolution or migration. We combined experiments and mathematical models to investigate how lag transiently alters the costs of CRISPR-Cas during a viral outbreak, and found that when viruses invade a primarily susceptible host population with a small sub-population of CRISPR-immune host even the CRISPR-immune cells face a large virus-induced reduction in fitness. Importantly, the costs of lag are only revealed by examining the dynamics of the system that occur immediately after viral challenge.

## 2. Models

### (a) Model framework

To model CRISPR-Cas immunity in a simple host-phage system, we built on classical host-phage chemostat (or 'virostat', see Discussion) models [43–46], where resources ($R$) are modelled explicitly as being supplied by some constant reservoir ($r_0$), and there is constant flow ($w$) of fresh media into the system (for a discussion of model construction and assumptions, see electronic supplementary material, Text S1). At the same time, the contents of the system (cells, viruses, resources) are removed at the same rate ($w$) in order to maintain a constant volume. For a detailed discussion of this class of models, see Weitz [46]. Our model consists of a system of ordinary differential equations with equations for resources, host and virus populations:

$$\overbrace{\dot{R}}^{\text{resources}} = \overbrace{w(r_0 - R)}^{\text{flow}} - \overbrace{\frac{evR}{z+R}(S+C)}^{\text{resource uptake by cells}}$$

$$\overbrace{\dot{S}}^{\text{susceptible host}} = \left( \overbrace{\frac{vR}{z+R}}^{\text{growth}} - \overbrace{\delta V}^{\text{infection}} - \overbrace{w}^{\text{flow}} \right) S$$

$$\overbrace{\dot{C}}^{\text{immune host}} = \left( \overbrace{\frac{vR}{z+R}}^{\text{growth}} - \overbrace{w}^{\text{flow}} \right) C + \overbrace{\mu\delta VS}^{\text{immunization}}$$

$$\overbrace{\dot{I}}^{\text{infected host}} = \overbrace{(1-\mu)\delta VS}^{\text{infection}} - \overbrace{\gamma I}^{\text{lysis}} - \overbrace{wI}^{\text{flow}}$$

$$\text{and} \quad \overbrace{\dot{V}}^{\text{viruses}} = \overbrace{w(v_0 - V)}^{\text{flow}} + \overbrace{\beta\gamma I}^{\text{lysis}} - \overbrace{\delta(S+C)V}^{\text{adsorption}}.$$

(2.1)

**Table 1.** Parameter and variable definitions for models discussed in main text.

| symbol | definition | value and units |
| --- | --- | --- |
| $R$ | resources | dynamic, µg ml$^{-1}$ |
| $S$ | susceptible host | dynamic, cells ml$^{-1}$ |
| $I$ | infected host | dynamic, cells ml$^{-1}$ |
| $C$ | CRISPR-immune host | dynamic, cells ml$^{-1}$ |
| $L$ | lagged host | dynamic, cells ml$^{-1}$ |
| $C_F$ | upregulated immune host | dynamic, cells ml$^{-1}$ |
| $M$ | surface mutant host | dynamic, cells ml$^{-1}$ |
| $V$ | viruses | dynamic, viruses ml$^{-1}$ |
| $e$ | resource conversion factor | $5 \times 10^{-7}$ µg cell$^{-1}$ [39,48] |
| $v$ | Max. growth rate | 2 h$^{-1}$ [51] |
| $z$ | half-saturation constant | 1 µg ml$^{-1}$ [39,48] |
| $r_0$ | resource concentration in reservoir | 350 µg ml$^{-1}$ [39,48] |
| $\kappa$ | growth cost of SM | 0.01 [30,31] |
| $w$ | flow rate | 0.3 h$^{-1}$ [39,48] |
| $\mu$ | probability of spacer acquisition | $10^{-7}$ [52] |
| $\delta$ | adsorption rate | $10^{-7}$ ml h$^{-1}$ [39,48] |
| $\beta$ | burst size | 80 viruses cell$^{-1}$ [52] |
| $\gamma$ | rate of lysis of infected cells | $\frac{4}{3}$ h$^{-1}$ [52] |
| $v_0$ | virus concentration in reservoir | 100 viruses ml$^{-1}$ |
| $\phi$ | recovery rate from immune lag | 10 h$^{-1}$ [29,30], 1000 h$^{-1}$ (estimated) |
| $\zeta$ | downregulation rate | 0.1 h$^{-1}$ |

Specifically, we equipped our host population with a CRISPR-Cas system, so that there is a population of naive, undefended-but-CRISPR-encoding host ($S$) that may become infected ($I$) by viruses ($V$). Each time a susceptible host is infected, it may undergo immunization with probability $\mu$ to become defended (i.e. spacer-possessing) host ($C$). This formulation is similar to other minimal models of CRISPR-Cas immunity (e.g. [39,47–50]). For some analyses, we also included a virus-resistant surface mutant (SM) strain in the model ($M$):

$$\overbrace{\dot{M}}^{\text{SM host}} = \left( \overbrace{\frac{(1-\kappa)vR}{z+R}}^{\text{growth}} - \overbrace{\frac{1}{w}}^{\text{flow}} \right) M,$$

(2.2)

with growth cost $\kappa$ associated with its surface mutation, and appropriate changes to our equation for resource dynamics. For a list of parameter and state variable definitions, see table 1 and for a discussion of model construction and assumptions, see the electronic supplementary material, Text S1. See the electronic supplementary material, figure S1 for a schematic of this base model.

Observe that in a small departure from the classical chemostat model we allow constant immigration of viruses into the system from some environmental pool ($v_0$). This is an entirely experimentally tractable modification (e.g. by adding set concentrations of virus to the resource reservoir), and better represents natural systems which are not closed and where hosts probably face constant challenges in the

form of newly arriving viruses. Note that this basic model only considers a single viral genotype, so that immune hosts will also be immune to immigrating viruses (though see outbreak simulations discussed later for simulations in which this is not the case). For traditional continuous culture without viral inflow simply let $v_0 = 0$.

Our model of CRISPR-Cas immunity is intentionally simple in that it neglects: (i) details of the spacer acquisition process, (ii) autoimmunity, (iii) spacer diversity, and (iv) viral coevolution. CRISPR-immune strains are modelled as a single, homogeneous pool of immune host ($C$) and viruses are not able to coevolve to overcome CRISPR-Cas immunity. Nevertheless, this model is a suitable scaffold on which to build a more complex model of immune lag. We provide detailed explorations of the spacer acquisition process in the electronic supplementary material, Text S2,S3 and figures S2–S4 (including acquisition from collapsed replication forks [52,53], acquisition from defective phages [54], and primed acquisition [10,11]), and show that the details of spacer acquisition are largely irrelevant for assessing the fitness cost of lag in a CRISPR-immune host. Furthermore, a careful analysis of autoimmunity, with rates estimated based on a realistic model of self versus non-self recognition (electronic supplementary material, Text S2), predicted that there should be essentially no impact of autoimmunity on the hosts' fitness (electronic supplementary material, figure S2), which is consistent with experimental efforts that have not detected any constitutive cost of CRISPR-Cas immunity [29–31]. We address spacer diversity and viral coevolution in our simulations of repeated viral outbreaks (see Results; electronic supplementary material, Text S8).

## (b) Immune lag

Meaden et al. [31] provide strong evidence that the inducible cost of CRISPR-Cas is associated with transient expression of viral genes and possible virus-induced reprogramming of cellular transcriptional networks. Because CRISPR-Cas immunity does not remove viral genetic material from the cell instantaneously, Meaden et al.'s [31] results suggest that even immune hosts face a temporary growth setback during infection while viruses transiently reprogramme the cell (figure 1). Virus-resistant surface mutants do not experience this growth setback, as viruses are unable to adsorb to the cell in the first place. We built a model of lagged CRISPR immunity (figure 1b). Consider the simple model described above. We added an equation for transiently infected but immune host ($L$):

$$\overset{\text{lagged}}{\overbrace{\dot{L}}} = \overset{\text{enter lag}}{\overbrace{\delta CV}} - \overset{\text{clearance}}{\overbrace{\phi L}} - \overset{\text{flow}}{\overbrace{wL}} , \qquad (2.3)$$

which are able to clear an infection at rate $\phi$ via cleavage of viral nucleotides, and modified the equation for immune host accordingly:

$$\dot{C} = \left( \overset{\text{growth}}{\overbrace{\frac{vR}{z+R}}} - \overset{\text{enter lag}}{\overbrace{\delta V}} - \overset{\text{flow}}{\overbrace{w}} \right) C$$
$$ + \overset{\text{immunization}}{\overbrace{\mu \delta VS}} + \overset{\text{clearance}}{\overbrace{\phi L}} . \qquad (2.4)$$

We also found that lag can be modelled in an even simpler four-parameter system with qualitatively similar results

(electronic supplementary material, Text S4 and figure S5). Thus, for completeness and comparison with experimental results, we present a parameter-rich model, but our results can be replicated with minimal models of host–phage interactions.

## (c) Upregulation of the CRISPR locus

The cas genes and CRISPR arrays of many hosts are transcriptionally upregulated in response to infection [55,56], or in situations where there is a high risk of infection (e.g. in a biofilm; [57–61]). The specific regulatory cues used by the CRISPR locus are diverse [56], and new methods are being developed to probe them [62]. Consider the case where the CRISPR locus is specifically upregulated in response to infection. The initial time-to-clearance of infection is unaffected by upregulation, but for some time after this first infection the host will be on 'high alert', producing many Cas proteins and crRNAs. We consider the case where this overproduction of CRISPR-Cas defence complexes allows the host to degrade viral genetic material before it can be expressed, thus avoiding any immune lag. We implemented this scenario (electronic supplementary material, figure S6) by letting recently immunized and lagged cells pass into a 'fast' immunity ($C_F$) state where the CRISPR-Cas system does not experience immune lag because the cas targeting genes are upregulated:

$$\dot{C}_F = \left( \overset{\text{growth}}{\overbrace{\frac{vR}{z+R}}} - \overset{\text{flow}}{\overbrace{w}} \right) C_F$$
$$ + \overset{\text{immunization}}{\overbrace{\mu \delta VS}} + \overset{\text{clearance}}{\overbrace{\phi L}} - \overset{\text{downregulation}}{\overbrace{\zeta C_F}} , \qquad (2.5)$$

and modified the equation for immune host accordingly:

$$\dot{C} = \left( \overset{\text{growth}}{\overbrace{\frac{vR}{z+R}}} - \overset{\text{enter lag}}{\overbrace{\delta V}} - \overset{\text{flow}}{\overbrace{w}} \right) C + \overset{\text{downregulation}}{\overbrace{\zeta C_F}} \qquad (2.6)$$

Note that we do not include any cost of increased transcription/translation in this model, as we have no empirical estimate or intuition for the scale of this cost, though one almost certainly exists (because in the absence of a cost the expression of CRISPR-Cas would be expected to be constitutively high). Also, observe that we modelled an upregulation of the cas targeting genes, which will reduce or eliminate immune lag (in our case eliminate), rather than the cas acquisition machinery, which would possibly increase autoimmunity (though both may be upregulated during infection because the cas genes are often transcribed as an operon). Finally, here we assume that cells return from the transcriptionally upregulated state ($C_F$) to the baseline state ($C$) at a constant rate ($\zeta$), though relaxing this assumption has little effect on the qualitative results of the model (electronic supplementary material, Text S5 and figure S7).

## 3. Results

### (a) A tipping point between CRISPR and surface mutant strategies at high viral titres

We found the equilibria of our lag model and determined their stability via linear stability analysis in order to characterize the ultimate outcome of competition between a laggy

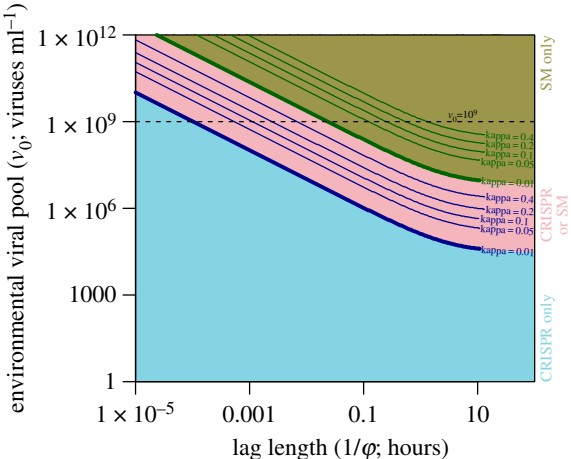

**Figure 2.** Model equilibria show a transition from complete reliance on CRISPR-Cas immunity to complete reliance on surface modifications (SM strain) by the host at very high viral titres. Shown are exact analytical solutions of model equilibria characterized via linear stability analysis (see the electronic supplementary material, Text S6). No parameter conditions permitting stable coexistence of the two host strains were observed. Lower blue region denotes region of parameter space in which the CRISPR-only equilibrium was the only stable equilibrium and upper brown region denotes the region in which the SM-only equilibrium was the only stable equilibrium. Middle pink region denotes the region in which both CRISPR-only and SM-only equilibria were stable, indicating that the final state of the system depends on initial conditions. Thick solid lines denote the boundaries corresponding to the different colour regions when the cost of the SM strains is low ($\kappa = 0.01$). Thin solid lines show how these boundaries shift towards larger values of $v_0$ for larger costs of the SM strain (higher $\kappa$). (Online version in colour.)

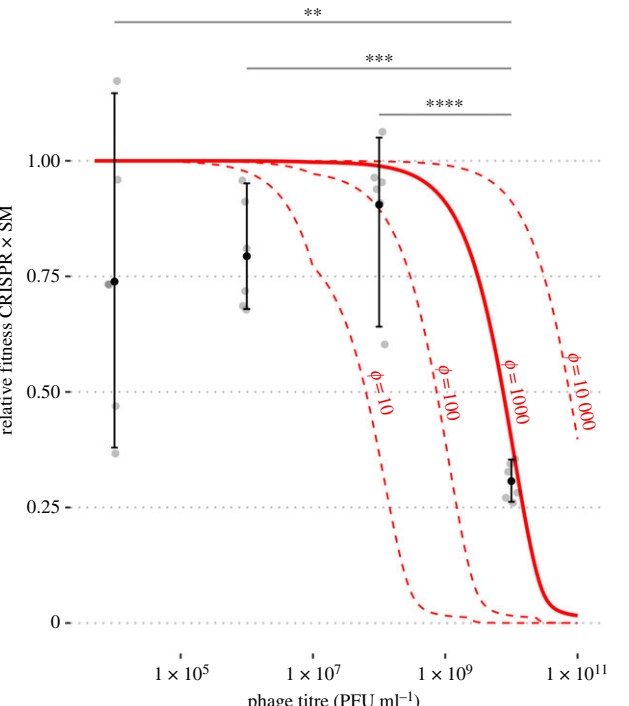

**Figure 3.** A strongly inducible cost of CRISPR-Cas immunity at high viral titre is consistent with 'laggy' CRISPR-Cas immunity. (*a*) Competition experiments between BIM2 and SM strains in the presence of various viral titres. Fitness calculated based on densities 1 day post infection. Solid red line shows results from numerical solutions of model with no cost for SM ($\kappa = 0$) and a short lag time ($\phi = 10^3$ h$^{-1}$, $1/\phi = 3.6$ s) for comparison. Asterisks indicate significant differences between conditions (**$p < 10^{-2}$; ***$p < 10^{-3}$; ****$p < 10^{-4}$; Tukey's honest significant difference test). These results qualitatively reflect the outcomes of similar previous experiments with the same strains (electronic supplementary material, figure S10; [29,30]), though that work suggests a much longer lag period ($\phi \approx 10$ h$^{-1}$, $1/\phi \approx 6$ min). (Online version in colour.)

CRISPR-immune strain and a costly SM strain (see the electronic supplementary material, Text S6 for equations describing equilibria and analysis details). Over a wide range of parameter values the model yielded a single stable CRISPR-immune equilibrium with an extinct SM strain (figure 2). Only when there was extremely high flow of viruses into the system (high $v_0$) did we see an alternative outcome where the only stable equilibrium was an SM-only state with an extinct CRISPR strain. For short lag times ($\phi \geq 10^3$ h$^{-1}$, ($1/\phi$) < 4 s), the 'tipping point' from an all-CRISPR to all-SM state occurred as the external viral pool ($v_0$) exceeded concentrations of $10^9$ PFU ml$^{-1}$ (figure 2). At intermediate lag times and viral densities, there was a region of bistability in which the initial conditions of the system determined whether or not it would end up in an SM- or CRISPR-only state, with both equilibria being stable. This bistability is a byproduct of CRISPR's ability to clear viruses from the environment and in doing so reduce the impact of lag when cells are at a high density (electronic supplementary material, S7 Text and figures S8,S9). In no case did the two strains, CRISPR and SM, coexist stably over the parameter regimes considered. Thus, our model predicts a sharp transition from a CRISPR strategy being favoured to an SM strategy being favoured at high viral titres. This is consistent with previous work on inducible immunity that saw a steep decrease in the relative fitness of a CRISPR-Cas immune strain when competed against an SM strain at very high viral titres [29].

However, previous experiments appear to disagree on the severity of the inducible cost of CRISPR-Cas immunity. In the original work on the topic, Westra *et al.* [29] observed a steep transition from high relative fitness (greater than 1) to a

relative fitness of essentially zero for a CRISPR strategy competed against an SM strategy in competition experiments with increasing multiplicity of infection (MOI), consistent with our model's predictions (electronic supplementary material, figure S10). More recently, Alseth *et al.* [30] did not observe this steep fitness decrease while performing nearly identical experiments. We suspected that these later experiments failed to capture the transition from high to low relative fitness seen in our model because they were not carried out to a sufficiently high viral titre. Therefore, we replicated the Alseth *et al.* [30] experiments with the same host-phage system, *P. aeruginosa* UCBPP-PA14 and its lytic phage DSM3*vir*, out to a higher viral titre ($10^{10}$ PFU ml$^{-1}$) and were able to capture the steep decrease in fitness of the CRISPR-immune strain at high MOI (figure 3), confirming our model predictions. We matched these competition experiments to model predictions by solving our model numerically (see the electronic supplementary material, Text S8 'Simulating competition experiments') to more precisely illustrate this point (figure 3, solid red line). Our lag model captures the major shift from CRISPR-to-SM strategy that happens at high viral densities as seen by Westra *et al.* [29], consistent with the idea that immune lag causes the inducible cost of CRISPR-Cas immunity (figure 3; electronic supplementary material, figure S10). Importantly, the original work by Westra *et al.* [29], showed

that the inducible cost of CRISPR-Cas immunity was not owing to virus-induced mortality, as even less-fit CRISPR-immune cells survived at high viral titres.

Finally, we note that while the qualitative results of these competition experiments are highly reproducible, with a steep decrease in the fitness of CRISPR-Cas immune strains occurring at high MOI, where exactly this transition occurs and the baseline relative fitness of the CRISPR-immune strain in the absence of virus appear to be quite variable between replicates and experiments (figure 3; electronic supplementary material, figure S10, and see figure S11 for a direct comparison between the two). Viruses and host cells were quantified using serial dilutions, introducing the possibility of multiplicative errors and perhaps making cross-experiment variability less surprising. This cross-experiment variability prevented us from obtaining precise lag estimates (we estimate that $10^{-3}$ h$^{-1} \leq 1/\phi \leq 0.1$ h$^{-1}$, in other words that $3.6$ s $\leq 1/\phi \leq 6$ min; electronic supplementary material, figure S11). Initial host density in particular can strongly affect model expectations (electronic supplementary material, figure S12). On the other hand, given large starting densities of host (approx. $10^7$ cells ml$^{-1}$), this variability is unlikely to arise from demographic stochasticity. For model results reported below, we include an analysis of both short ($\phi = 10^3$ h$^{-1}$, $1/\phi = 3.6$ s) and long ($\phi = 10$ h$^{-1}$, $1/\phi = 6$ min) lag times to capture the full range of experimental variability.

## (b) Immune lag is extremely costly during an outbreak of novel virus

Interestingly, the lags estimated in the last section are quite short, ranging from $3.6$ s ($\phi = 10^3$ h$^{-1}$) to $6$ min ($\phi = 10$ h$^{-1}$). This short length in part explains why such high viral titres (often greater than $10^9$ PFU ml$^{-1}$) are required to observe any effect of lag on the host population. The cost of lag seems to only be revealed when immune hosts are facing multiple subsequent viral infections. When will such high viral titres be achieved? We suspected that during a viral outbreak where only a small fraction of the host population is initially immune, viral titres might greatly exceed immune host densities and lead to a clear cost of lag.

We simulated an outbreak of 'novel' virus to which pre-existing CRISPR-Cas immunity did not exist in the population, or to which only be a very small proportion of the population was already immunized (see the electronic supplementary material, S8 'Simulating outbreaks'). In practice, this was achieved by initializing the system with a dense susceptible population ($10^8$ cells ml$^{-1}$) and a very small CRISPR immune population ($100$ cells ml$^{-1}$), both exposed to a small environmental viral pool ($v_0 = 100$ viruses ml$^{-1}$), and solving numerically. We found that during outbreaks of novel viruses immune lag can be extremely costly, leading to selection for an SM defence strategy over a CRISPR strategy, even when the SM strategy comes with a growth cost (figure 4a–c). The cost of lag was only apparent when we examined the early, transient dynamics of our model and is relevant to natural systems where outbreaks of novel or mutant viral strains may occur at moderate to high frequency. Unlike our results described above for systems at equilibrium, even a very low rate of immigration of novel viruses into the system can lead to a massive reduction in the fitness of a CRISPR-Cas relative to an SM strategy early on if most hosts are not already immunized.

During our simulated outbreaks, as the viral population spikes early it suppresses the initial growth of the CRISPR-immune host, leading the SM population to dominate (figure 4a–c; electronic supplementary material, figure S13). This initial dominance of SM even occurs when the cost of an SM strategy is very high, up to a 20% cost with a short lag and well over a 40% cost with a long lag (electronic supplementary material, figure S14). The only way for the CRISPR-immune host to dominate during an outbreak is for the duration of immune lag ($1/\phi$) to approach zero (figure 4a), but even short lags ($\phi = 10^3$ h$^{-1}$, $1/\phi = 3.6$ s) result in a substantial initial expansion of the SM population (figure 4b).

Even though an SM strategy will dominate immediately after an outbreak, if the SM strategy is sufficiently costly, then the system will eventually return to a CRISPR-dominated equilibrium. How long will this return to CRISPR-dominance take, and what happens if the system is perturbed again before then? In natural communities, novel viral strains to which the host lacks pre-existing immunity may emerge via mutation or immigrate into the system via dispersal. We found that even moderately frequent outbreaks can lead to selection against a CRISPR-based defence strategy.

We simulated our system's dynamics under repeated outbreaks at set intervals (see the electronic supplementary material, S8 'Simulating intermittent outbreaks'), corresponding to either the emergence of an escape mutant in the viral population or the arrival of a novel viral species into the system against which the host lacks a pre-existing spacer. We observe that for long lags ($\phi = 0.1$ h$^{-1}$, $1/\phi = 6$ min), if outbreaks occur even with moderate (monthly) frequency, immune lag will prevent a CRISPR strategy from rising to dominance (figure 4d). These results agree with empirical observations that the repeated addition of susceptible host into a host-phage system promotes the evolution of an SM strain over a CRISPR-immune strain [32]. Note that we assume that outbreaks affect both CRISPR and SM strains, so that novel virus can overcome both defence strategies, in order to compare strategies on an equal footing and calculate the precise cost of lag. In reality, the probability of a viral mutant escaping spacer targeting versus the probability of a viral mutant being able to target a new or modified host receptor are likely to be quite different, which would alter the frequency of outbreaks that host populations employing these two strategies would experience. In all likelihood, the respective rates of coevolutionary dynamics for hosts with CRISPR and SM strategies will vary a great deal across systems in complex ways that are difficult to capture with simple models.

CRISPR-Cas systems have one important advantage that our model neglects—different hosts may have different spacers, leading to a great deal of immune diversity in the population. This diversity is protective, as it makes it much more difficult for viral escape mutants to gain a foothold [63–65]. To account for host immune diversity, we varied the fraction of the host population susceptible to each novel viral outbreak (figure 4e). Even daily outbreaks affecting less than 10% of the host population will prevent the dominance of a CRISPR-Cas defence strategy when lags are long ($\phi = 10$ h$^{-1}$, $1/\phi = 6$ min). The spacer frequency distribution in host populations is often highly skewed, so that a few spacers are found among a large fraction of hosts [66–68]. Given that viral outbreaks are generally expected to affect the most abundant host sub-populations [69], in the presence of small, frequent outbreaks, such a skewed distribution of

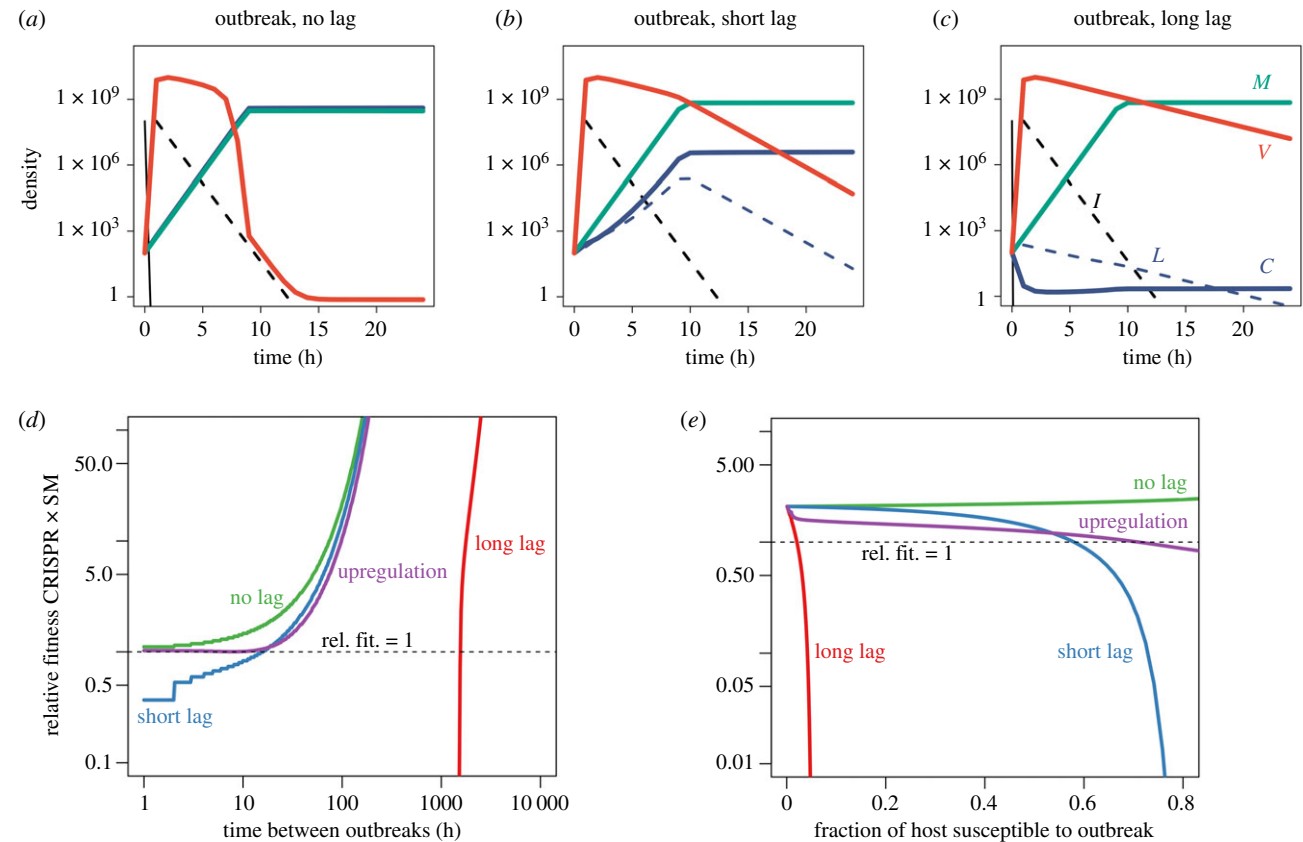

**Figure 4.** Immune lag prevents the dominance of a CRISPR-immune (*C*) strategy over a costly SM strategy (*M*) during repeated outbreaks. (*a–c*) Immune lag prevents CRISPR-Cas from out-competing a costly SM strategy ($\kappa = 0.01$) early on in an outbreak (starting from a dense susceptible population, $S = 10^8$ cells ml$^{-1}$). Numerical solutions for lag model with CRISPR-immune (purple), SM (green), viral (orange) and susceptible (black) populations shown. Note that as the system approaches equilibrium the CRISPR strain will eventually out-compete the SM population (though this may take a long time), regardless of lag. (*d,e*) Immune lag prevents the CRISPR strategy from out-competing an SM strategy when outbreaks are frequent. Results shown for iterated outbreak model with the size and frequency of outbreaks varied. Outbreaks of novel virus must be rare or affect a small fraction of the community for the CRISPR strategy to out-compete an SM strategy. 'Short' and 'long' lags correspond to the upper ($\phi = 10^3$ h$^{-1}$, $1/\phi = 3.6$ s) and lower bounds ($\phi = 10$ h$^{-1}$, $1/\phi = 6$ min) estimated for $\phi$ in figure 3 and the electronic supplementary material, figure S10. 'Upregulation' refers to a system with a CRISPR-immune strain with a long lag ($\phi = 10$ h$^{-1}$, $1/\phi = 6$ min), but that can be upregulated to a 'fast' immune state ($\zeta = 0.1$ h$^{-1}$). In (*e*) outbreaks affect 50% of the host population (interval between outbreaks varied) and in (*d*) outbreaks occur every 24 h (fraction of host population affected varied). (Online version in colour.)

immune variants would make it very difficult for a lagged CRISPR-immune host to out-compete an SM population.

## (c) Inducible defences can mitigate the effects of immune lag

The *cas* operon, or a subset of *cas* genes, are often transcriptionally upregulated in response to infection or to conditions that indicate a high risk of infection [56]. We found that if strong upregulation occurs after infection, so that the resulting 'fast immune' cells with an upregulated CRISPR locus do not experience lag, the overall effects of immune lag can largely be mitigated during an outbreak (figure 4*d,e*). This result is relatively robust to variations in the rate at which cells return to normal expression levels ($\zeta$), though high rates of return will ameliorate lag less than lower rates (electronic supplementary material, figure S15).

## (d) Laggy immunity is less costly in slow growing hosts

It has been suggested that slow growing and less dense host populations will favour CRISPR-Cas immunity, whereas fast growing, dense populations will favour alternative defence strategies (e.g. SM; [27,28]). Consistent with growth affecting

the evolution of host defence strategy, Westra *et al.* [29] showed that high resource environments favoured an SM strategy over a CRISPR strategy in direct competition experiments. We wondered if lag could partially explain this phenomenon. Using our lag model, we found that in host populations with a high maximal growth rate the cost of immune lag was much greater than in slower growing populations (electronic supplementary material, figure S16). Thus, if CRISPR-Cas immunity is laggy, it is much more likely to be favoured over an SM strategy if the host has a slow maximal growth rate. The opposite is true when CRISPR-Cas has no lag (electronic supplementary material, figure S16). Intuitively, the temporary slow down in growth for hosts with a laggy immune systems is felt less strongly if growth is already slow, whereas the impact is much greater in a highly competitive system with fast growers.

## 4. Discussion

We built a biologically motivated model of CRISPR-Cas immunity that links population-scale host–virus dynamics to molecular-scale changes within the cell. In doing so, we were able to demonstrate that immune lag can strongly impact the evolution of immune strategy in some prokaryotes

[29,31]. Immune lag's effect is felt most severely during an outbreak of novel virus. We showed that for even moderately frequent outbreaks of novel viruses that are unrecognized by the CRISPR-Cas immune system, immune lag will lead to selection against CRISPR-Cas in favour of other defence strategies (e.g. surface modifications). Note that we are making an argument here about selective forces acting on immune strategy—the rate and mechanics of how CRISPR-Cas functionality might be lost in natural settings, as well as the implications of this loss, are outside the scope of this study and have been explored elsewhere [22,39].

Even considering the beneficial effects of priming seen in many systems [10,11], where partial spacer-protospacer matches stimulate rapid spacer acquisition thus allowing hosts to 'update' their immune memory against viral escape mutants, it is not unreasonable to expect wholly novel outbreaks on a daily or weekly timescale for natural systems with high viral migration. That being said, primed adaptation can still help overcome short lags in the special case where outbreaks of novel virus affect the entire population of defended hosts (electronic supplementary material, figure S4). Our results emphasize the benefits of having multiple redundant spacers towards the same target, as even temporary loss of immunity in a subset of the population can lead to strongly negative fitness effects for the entire immune host population owing to lag induced by the resulting high-density viral bloom.

Immune lag has strong negative fitness effects even when CRISPR-immune strains are competed against very costly SM strains. In culture, SM strategies may occasionally be essentially cost-free, but in natural systems surface molecules that act as viral receptors often play an important role in host fitness, which can prevent the emergence of SM strains [30,70]. Nevertheless, during an outbreak in a primarily susceptible population very costly SM strains still rise to dominance (electronic supplementary material, figure S14). Thus, lag is likely to be relevant even in natural systems where surface modifications are very costly. Phage DMS3vir uses the host pilus as a receptor, meaning that SM mutants are defective in terms of motility, though we saw no great difference in fitness between SM and CRISPR strains in the absence of phage in our experiments (figure 3; [29–31]).

Some immune host strains may be able to partially avoid the effects of lag. The impact of lag can be mitigated by transcriptional upregulation of the *cas* locus (figure 4). Thus, lag may help explain why expression of the *cas* genes is tightly regulated in many systems [30,31], in combination with other explanations such as avoidance of autoimmunity [71]. Additionally, lag seems to have less of an impact on slow growing host populations, perhaps explaining in part the suggested, though not yet systematically demonstrated, pattern in which CRISPR-Cas is more common among slow growing and low density taxa ([27,28]; though of course many organisms capable of fast growth, including *P. aeruginosa*, also have CRISPR-Cas). These variations may perhaps explain why the prevalence of CRISPR-Cas immunity varies so widely between different groups of organisms (e.g. between anaerobes and aerobes [72], between bacteria and archaea [8,73]).

We emphasize that expression of viral genes and/or changes in host expression are not the only phenomena that could lead to a slowdown in the growth of immune host upon infection. For example, membrane depolarization owing to viral injection could lead to a transient growth slowdown. Our model is agnostic to the mechanism causing lag, and only requires that two criteria be satisfied: (i) the virus-induced fitness reduction is owing to growth inhibition rather than increased mortality (based on Westra *et al.* [29]), and (ii) the virus-induced fitness reduction is felt only by cells that allow for viral entry (i.e. not SM cells). That being said, in the experimental system we consider, Meaden *et al.* [31] provide strong evidence that the transient expression of viral genes leads to a reduction in the fitness of immune host.

Similar to the uncertainty surrounding the mechanisms causing lag, we do not have a good estimate for the length of the lag period. Our estimated lag durations from different experiments ranged over two orders of magnitude (from seconds to minutes; figure 3; electronic supplementary material, figures S10, S11), even though the same strains were used across experiments [29,30]. The competition experiments we used to parametrize our model probably lack the precision for an accurate estimation of lag duration (electronic supplementary material, figure S12). Alternative experimental approaches that more directly assess growth slowdowns (e.g. single-cell analyses using microfluidic devices [74]) may be required to obtain accurate parameter estimates. We analysed the population-level implications of immune lag, but much is left to be done in order to characterize the cellular-level mechanisms and effects of lag.

Finally, it is clear that the severity of immune lag is a shared trait determined by both host and virus. Different viruses reprogramme the cell in different ways and to different degrees (e.g. [35]). Similarly, hosts will probably vary in their susceptibility to reprogramming. Our overall conclusions are relatively insensitive to this variability, as even short lags can have a severe impact on host fitness during an outbreak (electronic supplementary material, figure S14). Nevertheless, this variability makes it difficult to put realistic bounds on our lag parameters, and as we show in figure 4, upregulation of the CRISPR locus may mitigate immune lag for the host [56]. For other types of intracellularly acting defence, such as abortive infection systems in which infected host cells do not recover, lag may be irrelevant. In any case, we highlight an important parameter of virus–host dynamics, the recovery rate of defended cells to viral infection ($\phi$), that is not typically measured or considered in theoretical treatments of host-phage systems. This parameter is probably universal to all intracellularly acting DNA- or RNA-degrading defence systems, including restriction–modification systems, which are nearly ubiquitous [75]. Immune lag is quite possibly a widespread phenomenon common to many classes of defence systems acting within the cell (e.g. [76–78]), and deserves consideration in any population-level study of prokaryotic antiviral defences.

Data accessibility. All code and raw data necessary to run models and generate figures are available at https://github.com/jlw-ecoevo/immunelag.

Authors' contributions. J.L.W. conceptualization, data curation, formal analysis, funding acquisition, investigation, methodology, project administration, software, supervision, validation, visualization, writing—original draft, writing—review and editing; E.O.A. data curation, investigation, resources, writing—review and editing; S.M. conceptualization, writing—review and editing; E.R.W. conceptualization, data curation, investigation, resources, writing—review and editing; J.A.F. conceptualization, funding acquisition, project administration, supervision, writing—review and editing. All authors gave final approval for publication and agreed to be held accountable for the work performed therein.

Competing interests. We declare we have no competing interests.

Funding. J.L.W. was supported by a postdoctoral fellowship in marine microbial ecology from the Simons Foundation (award no. 653212).

E.R.W. was supported by grants from the European Research Council (ERC-STG-2016-714478-EVOIM MECH), the Biotechnology and Biological Sciences Research Council (BB/N017412/1), and the Natural Environment Research Council (NE/M01835/1). J.A.F. was supported by the Simons Foundation Collaboration on Computational Biogeochemical Modeling of Marine Ecosystems (CBIOMES) grant 549943.

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
