## [Peer Review File · Proceedings of the Royal Society B: Biological Sciences]

Review History

RSPB-2021-0435.R0 (Original submission)

Review form: Reviewer 1

Recommendation

Accept with minor revision (please list in comments)

Scientific importance: Is the manuscript an original and important contribution to its field?
Excellent

General interest: Is the paper of sufficient general interest?
Good

Quality of the paper: Is the overall quality of the paper suitable?
Good

Is the length of the paper justified?
Yes

Should the paper be seen by a specialist statistical reviewer?
No

Do you have any concerns about statistical analyses in this paper? If so, please specify them explicitly in your report.

No

It is a condition of publication that authors make their supporting data, code and materials available - either as supplementary material or hosted in an external repository. Please rate, if applicable, the supporting data on the following criteria.

Is it accessible?

Yes

Is it clear?

Yes

Is it adequate?

Yes

Do you have any ethical concerns with this paper?

No

Comments to the Author

The manuscript considers the question of whether a lag in immune response may be partly responsible for experimental evidence that CRSIPs-Cas systems are absent from many host organisms. They develop mathematical models to test their hypothesis, guided by experimental data. They find that indeed a lag in immune response can make CRISPR systems uncompetitive against a non-laggy mutant, even when that mutant carries a cost in reduced growth. The effect is particularly strong for slow-growing and small populations.

The manuscript addresses an interesting question, and the model is used to good effect to make their key arguments. The writing is pretty good - I would say that as a modeller I found it ok, but I wonder if the model construction could do with more detail/motivation for a biological audience. But overall a really nice manuscript that got me thinking.

Despite the insistence that the model is 'simple' - and perhaps it is all that is minimally needed - I think it is actually quite complex to unpick. The more detailed models in the SI are particularly so. I don't really have a major criticism at all - I think this is a very nice piece of work. I guess the most important comment I have is that, as with all models, there are lots of assumptions here and I feel that they could be covered better. Doing so in the SI would probably be fine, but just details like why you chose the chemostat set-up and what that might mean for the results, why you chose the particular forms of growth, infection, downregulation, etc.

Other than that I just have some thoughts and suggestions that occurred to me while reading the manuscript:

P3 L74 (the model) - I have a couple of questions about the model set-up:

* With your chemostat assumption, a constant volume is only preserved if you can be sure that the total 'outgoing' quantities equal the total 'incoming' quantities. It's not immediately clear that this is true. The rates, w , may be the same, but of course if your incoming quantity is $w \cdot (r_0 + v_0)$ and your outgoing quantity is $w \cdot (R + S + C + I + V)$ these may well be different. It's not even clear to me they are all measured with the same units.

* COuld you move your paragraph about v_0 higher up - I actually wrote a comment asking about this before finding the relevant text a little later.

* Am I right that you assume a proportion, μ , of infection events actually result in immunisation, and the rest full infection? This is presumably similar to immune priming as seen in plants/insects?

P6 L142 - Can you comment more on your choice of downregulation function as this Holling/Hill type function. It's an interesting form with the 'force of infection' term giving the half-rate threshold. You could perhaps include a couple of plots (in the SI if needed).

P7 L145 - At the start of the results - and in the figure caption for figure 1 - please could you define SM again?

P7 L146 - The cost here is pretty small - 1% drop in growth? I find it interesting how unlikely it is to win the competition. What's your thinking on this? Presumably the loss due to both lysis and lost growth while infected is not that great? Connected to this, you don't give the parameter values used anywhere in the main text, but I think they should be here rather than just in the SI.

P7 L156 - As a modeller I'm interested in how the SM equilibrium obviously appears as a saddle-node equilibrium, giving rise to the bistable region. This will be to do with the Holling/Hill-type growth rates I guess. Perhaps it is not so interesting biologically though!

P8 L168 - Just to be clear, by 'in silico' do you just mean you ran your model for these different titres? If so, how were you measuring fitness here? I'm not going to insist on you doing the work, but it would have been interesting to set it up as a stochastic model and see if the variation ends up anything like from the experiments.

P8 L190 - I find 'non-equilibrium' a bit vague, as it could be that the underlying model cycles (given the free-living viruses and non-linear growth rates it would be possible here) and that would also be termed non-equilibrium. Maybe specify you mean during transient growth or early stages or something?

P8 L200 - I know the full result is in the SI, but could you state in the main text what the value of kappa is? Is a 10% growth loss very high? Do we have a benchmark to judge that?

P10 Figure 4 - When plotting here, while C is the only cell type to have defences, you can define the CRISPR-type as being C+CF+L couldn't you? I'm guessing this doesn't make much difference - it looks like L decreases fairly rapidly, and presumably as V decreases so does CF, I just wondered what these total numbers looked like?

Review form: Reviewer 2

Recommendation

Reject - article is scientifically unsound

Scientific importance: Is the manuscript an original and important contribution to its field?

Good

General interest: Is the paper of sufficient general interest?

Good

Quality of the paper: Is the overall quality of the paper suitable?

Marginal

Is the length of the paper justified?

Yes

Should the paper be seen by a specialist statistical reviewer?

No

Do you have any concerns about statistical analyses in this paper? If so, please specify them explicitly in your report.

No

It is a condition of publication that authors make their supporting data, code and materials available - either as supplementary material or hosted in an external repository. Please rate, if applicable, the supporting data on the following criteria.

Is it accessible?

Yes

Is it clear?

Yes

Is it adequate?

Yes

Do you have any ethical concerns with this paper?

No

Comments to the Author

Overall:

This is an interesting paper on the relationship between the strength of viral pressure and the relative benefits of CRISPR-Cas immunity in light of 'lags' (i.e., the time it takes before a system can clear an infection). In doing so, the team assumes immunity is 100% effective in the long-run (in the absence of other mortality factors), but that there is a cost: the time to clear can become excessive in the limit of relatively high infection rates and/or long lags. This becomes an effective growth cost which can be compared against the costs of surface mutations. The model formalism is largely solid, but there are a few major issues.

First, in the absence of parameters, any mathematical analysis, the team has missed an opportunity to explain the self-evident tradeoff between viral pressure (v_0) and lag time ($1/\phi$). Doing so would help to explain the simulation results. I believe that the 'mathematical' results may in fact be contained in the SI, but the paper does not make explicit how they estimated the break points between 'phases'.

Second, the extension of the model to include upregulation is mis-specified, i.e., it is wrong. The denominator of the 'downregulation' term adds up a rate with a density. That is not possible. Now perhaps there is some typo or reason for this error, but in my view, this model extension and results are incorrectly specified. I have cross-checked Table 1 (which itself misses multiple units and mis-specifies the units of the adsorption rate).

Finally, the key (and brief) result is simply that when there is lag, then high viral pressure means that CRISPR-Cas strains are worse off than surface mutation strains - but that this ultimately depends on inherent costs of growth, lag time, and viral intensity. Hence, the exciting direct piece is a link between theory and figure 3 (a new experiment). Yet, for all the talk of lag, the data seems to be explained with basically 0 lag (if I assume that the units are hrs, then 10^3 for ϕ implies 1/1000 of an hour lag, or essential 0 lag). If that is true, then how important is lag really? It could also be that CRISPR immunity is not 100% perfect, and at these high levels, then such imperfect defense could also lead to a fitness difference. There are multiple leaps of faith to explain data that doesn't fit the model and in fact the bulk of the SI includes an analysis of a different more complex model.

In sum: the core idea is compelling, but the gap between model and experiment and mis-specification of a core component detracts (and indeed, the error in part of the model renders part

of the results unreviewable at this stage).

Major Revision:

Model framework – Part 1 – Presentation

The overall structure is sensible and well described. But, improvements would be welcome. Figure 1 is nice, but also misses a key point -> the advantage of surface mutations are evident when there is no cost to such mutations, otherwise, even rapid reproduction may be slower than the lag. Also, title should be 'Lagged immunity' and not 'Laggy Immunity'. Hence, note that the growth cost parameter (κ) in equation 2 would imply alternative cases for the right hand side of Figure 1, one in which many more cells were made and one in which very few were made. Note that the first part of the results assumes $\kappa = 0.01$, implying that it is critical to include two versions of the κ parameter in the schematic. A schematic should also include a standard compartmental model view of the dynamics, especially including the addition of Eqs. 3 and 4 that imply different variants of the framework.

Model framework – Part 2 – Misspecification of the upregulation component

Indeed, I am confused by the meaning of equations (5) and (6). This variant of the model seems to retain a lagged state, which then goes into a fast immunity state and then a slow immunity state. But, strangely enough, in Equation (6) the immune cells seem just as susceptible as susceptible cells (w/out CRISPR-Cas immunity). In addition, Equations (5) and (6) are mis-specified and have a technical error. Note that in the effort to downregulate the downregulation rate, the team uses a denominator that adds up two variables with different units. C_f has units of density, whereas ΔV has units of inverse time. These two values can't be added in this way – it's simply a mis-specified model equation. If the team used this particular setting, then any/all results using this model equation must be revised. If this is a typo, then the team should fix it, but my feeling here is that the model is in fact mis-specified. As stated, all model results with upregulation of the CRISPR locus have a mis-specified model (e.g., later Results and Figure 5).

Result description

SM vs. Lagged CRISPR

The presentation of Figure 2 w/out dynamics precludes understanding. In addition, a choice of a virostate model is unusual. There is also no table of parameters. And, I fully expect that the transition line will change w/changes in κ . Note that the effective growth cost here is not analyzed. That is a missed opportunity. SM hosts have a growth cost of κ . Whereas, when $V=v_0$, then there is an intrinsic switch to lagged states (ΔV_0) and then an intrinsic rate of return (ϕ). Why did the authors not try and combine these to construct a simplified theory for the transition (which clearly must have lag length be small when V_0 is high to avoid the continual push into the lagged state leading to a severely depressed growth). The authors say 'analyzed our model', but I don't know if that's mathematical analysis or computation. I don't see any math, so I can only presume they did not in fact come up with the theory that seems evident from the nature of the transitions. The transition lines seem jagged at times, so I am not sure what this figure actually represents. There are also no units on Figure 2 (unacceptable, throughout the use of units must be included if the team is going to compare across data/models, what is 10^3 for ϕ exactly in terms of a rate). Now, having reviewed Table S1 my understanding is that the authors use a short lag time of $\phi = 10^3$. That would seem to correspond to an average lag time of 1/1000 hrs or approximately 3 seconds. What does that mean biologically? For a model that is trying to explain an outcome w/lag to find that the lag is ~ 3 seconds implies that perhaps lag is not the answer. This is problematic.

Novel outbreaks -> This section is intriguing, but the invocation of novel viruses implies that the actual system used is highly diverse, and the models no longer tell the reader what is happening. Note that because the team uses multiple models, each time they invoke a 'simulation' they need to tell the readers which system of equations they are using. This makes it very hard to parse the

results.

Next, Figure 3 is quite interesting, but it deserves its own section.

L81: it's -> its

L273: host -> hosts

No table of parameters w/units.

Decision letter (RSPB-2021-0435.R0)

09-Apr-2021

Dear Dr Weissman:

I am writing to inform you that your manuscript RSPB-2021-0435 entitled "Immune Lag Is a Major Cost of Prokaryotic Adaptive Immunity During Viral Outbreaks" has, in its current form, been rejected for publication in Proceedings B.

This action has been taken on the advice of referees, who have recommended that substantial and fundamental revisions are necessary. With this in mind we would be happy to consider a resubmission, provided the comments of the referees are fully addressed. However please note that this is not a provisional acceptance.

Sincerely,

Professor Hans Heesterbeek

Associate Editor

Board Member: 1

Comments to Author:

Your interesting manuscript has been assessed by two expert reviewers. Both found the ideas compelling and potentially interesting. However, each raises some criticisms about the model construction and interpretation. Of particular concern is the second major issue raised by reviewer 2 i.e. that the model extension is fundamentally wrong.

Reviewer(s)' Comments to Author:

Referee: 1

Comments to the Author(s)

The manuscript considers the question of whether a lag in immune response may be partly responsible for experimental evidence that CRSIP5-Cas systems are absent from many host organisms. They develop mathematical models to test their hypothesis, guided by experimental data. They find that indeed a lag in immune response can make CRISPR systems uncompetitive against a non-laggy mutant, even when that mutant carries a cost in reduced growth. The effect is particularly strong for slow-growing and small populations.

The manuscript addresses an interesting question, and the model is used to good effect to make their key arguments. The writing is pretty good - I would say that as a modeller I found it ok, but I wonder if the model construction could do with more detail/motivation for a biological audience. But overall a really nice manuscript that got me thinking.

Despite the insistence that the model is 'simple' - and perhaps it is all that is minimally needed - I think it is actually quite complex to unpick. The more detailed models in the SI are particularly so. I don't really have a major criticism at all - I think this is a very nice piece of work. I guess the most important comment I have is that, as with all models, there are lots of assumptions here and I feel that they could be covered better. Doing so in the SI would probably be fine, but just details like why you chose the chemostat set-up and what that might mean for the results, why you chose the particular forms of growth, infection, downregulation, etc.

Other than that I just have some thoughts and suggestions that occurred to me while reading the manuscript:

P3 L74 (the model) - I have a couple of questions about the model set-up:

* With your chemostat assumption, a constant volume is only preserved if you can be sure that the total 'outgoing' quantities equal the total 'incoming' quantities. It's not immediately clear that this is true. The rates, w , may be the same, but of course if your incoming quantity is $w \cdot (r_0 + v_0)$ and your outgoing quantity is $w \cdot (R + S + C + I + V)$ these may well be different. It's not even clear to me they are all measured with the same units.

* COuld you move your paragraph about v_0 higher up - I actually wrote a comment asking about this before finding the relevant text a little later.

* Am I right that you assume a proportion, μ , of infection events actually result in immunisation, and the rest full infection? This is presumably similar to immune priming as seen in plants/insects?

P6 L142 - Can you comment more on your choice of downregulation function as this Holling/Hill type function. It's an interesting form with the 'force of infection' term giving the half-rate threshold. You could perhaps include a couple of plots (in the SI if needed).

P7 L145 - At the start of the results - and in the figure caption for figure 1 - please could you define SM again?

P7 L146 - The cost here is pretty small - 1% drop in growth? I find it interesting how unlikely it is to win the competition. What's your thinking on this? Presumably the loss due to both lysis and

lost growth while infected is not that great? Connected to this, you don't give the parameter values used anywhere in the main text, but I think they should be here rather than just in the SI.

P7 L156 - As a modeller I'm interested in how the SM equilibrium obviously appears as a saddle-node equilibrium, giving rise to the bistable region. This will be to do with the Holling/Hill-type growth rates I guess. Perhaps it is not so interesting biologically though!

P8 L168 - Just to be clear, by 'in silico' do you just mean you ran your model for these different titres? If so, how were you measuring fitness here? I'm not going to insist on you doing the work, but it would have been interesting to set it up as a stochastic model and see if the variation ends up anything like from the experiments.

P8 L190 - I find 'non-equilibrium' a bit vague, as it could be that the underlying model cycles (given the free-living viruses and non-linear growth rates it would be possible here) and that would also be termed non-equilibrium. Maybe specify you mean during transient growth or early stages or something?

P8 L200 - I know the full result is in the SI, but could you state in the main text what the value of κ is? Is a 10% growth loss very high? Do we have a benchmark to judge that?

P10 Figure 4 - When plotting here, while C is the only cell type to have defences, you can define the CRISPR-type as being $C+CF+L$ couldn't you? I'm guessing this doesn't make much difference - it looks like L decreases fairly rapidly, and presumably as V decreases so does CF, I just wondered what these total numbers looked like?

Referee: 2

Comments to the Author(s)

Overall:

This is an interesting paper on the relationship between the strength of viral pressure and the relative benefits of CRISPR-Cas immunity in light of 'lags' (i.e., the time it takes before a system can clear an infection). In doing so, the team assumes immunity is 100% effective in the long-run (in the absence of other mortality factors), but that there is a cost: the time to clear can become excessive in the limit of relatively high infection rates and/or long lags. This becomes an effective growth cost which can be compared against the costs of surface mutations. The model formalism is largely solid, but there are a few major issues.

First, in the absence of parameters, any mathematical analysis, the team has missed an opportunity to explain the self-evident tradeoff between viral pressure (v_0) and lag time ($1/\phi$). Doing so would help to explain the simulation results. I believe that the 'mathematical' results may in fact be contained in the SI, but the paper does not make explicit how they estimated the break points between 'phases'.

Second, the extension of the model to include upregulation is mis-specified, i.e., it is wrong. The denominator of the 'downregulation' term adds up a rate with a density. That is not possible. Now perhaps there is some typo or reason for this error, but in my view, this model extension and results are incorrectly specified. I have cross-checked Table 1 (which itself misses multiple units and mis-specifies the units of the adsorption rate).

Finally, the key (and brief) result is simply that when there is lag, then high viral pressure means that CRISPR-Cas strains are worse off than surface mutation strains - but that this ultimately depends on inherent costs of growth, lag time, and viral intensity. Hence, the exciting direct piece is a link between theory and figure 3 (a new experiment). Yet, for all the talk of lag, the data seems to be explained with basically 0 lag (if I assume that the units are hrs, then 10^3 for ϕ implies 1/1000 of an hour lag, or essential 0 lag). If that is true, then how important is lag really? It could also be that CRISPR immunity is not 100% perfect, and at these high levels, then such imperfect defense could also lead to a fitness difference. There are multiple leaps of faith to

explain data that doesn't fit the model and in fact the bulk of the SI includes an analysis of a different more complex model.

In sum: the core idea is compelling, but the gap between model and experiment and mis-specification of a core component detracts (and indeed, the error in part of the model renders part of the results unreviewable at this stage).

Major Revision:

Model framework – Part 1 – Presentation

The overall structure is sensible and well described. But, improvements would be welcome.

Figure 1 is nice, but also misses a key point -> the advantage of surface mutations are evident when there is no cost to such mutations, otherwise, even rapid reproduction may be slower than the lag. Also, title should be 'Lagged immunity' and not 'Laggy Immunity'. Hence, note that the growth cost parameter (κ) in equation 2 would imply alternative cases for the right hand side of Figure 1, one in which many more cells were made and one in which very few were made.

Note that the first part of the results assumes $\kappa = 0.01$, implying that it is critical to include two versions of the κ parameter in the schematic. A schematic should also include a standard compartmental model view of the dynamics, especially including the addition of Eqs. 3 and 4 that imply different variants of the framework.

Model framework – Part 2 – Misspecification of the upregulation component

Indeed, I am confused by the meaning of equations (5) and (6). This variant of the model seems to retain a lagged state, which then goes into a fast immunity state and then a slow immunity state. But, strangely enough, in Equation (6) the immune cells seem just as susceptible as susceptible cells (w/out CRISPR-Cas immunity). In addition, Equations (5) and (6) are mis-specified and have a technical error. Note that in the effort to downregulate the downregulation rate, the team uses a denominator that adds up two variables with different units. C_f has units of density, whereas ΔV has units of inverse time. These two values can't be added in this way – it's simply a mis-specified model equation. If the team used this particular setting, then any/all results using this model equation must be revised. If this is a typo, then the team should fix it, but my feeling here is that the model is in fact mis-specified. As stated, all model results with upregulation of the CRISPR locus have a mis-specified model (e.g., later Results and Figure 5).

Result description

SM vs. Lagged CRISPR

The presentation of Figure 2 w/out dynamics precludes understanding. In addition, a choice of a virostate model is unusual. There is also no table of parameters. And, I fully expect that the transition line will change w/changes in κ . Note that the effective growth cost here is not analyzed. That is a missed opportunity. SM hosts have a growth cost of κ . Whereas, when $V=v_0$, then there is an intrinsic switch to lagged states (ΔV_0) and then an intrinsic rate of return (ϕ). Why did the authors not try and combine these to construct a simplified theory for the transition (which clearly must have lag length be small when V_0 is high to avoid the continual push into the lagged state leading to a severely depressed growth). The authors say 'analyzed our model', but I don't know if that's mathematical analysis or computation. I don't see any math, so I can only presume they did not in fact come up with the theory that seems evident from the nature of the transitions. The transition lines seem jagged at times, so I am not sure what this figure actually represents. There are also no units on Figure 2 (unacceptable, throughout the use of units must be included if the team is going to compare across data/models, what is 10^3 for ϕ exactly in terms of a rate). Now, having reviewed Table S1 my understanding is that the authors use a short lag time of $\phi = 10^3$. That would seem to correspond to an average lag time of 1/1000 hrs or approximately 3 seconds. What does that mean biologically? For a model that is trying to explain an outcome w/lag to find that the lag is ~3 seconds implies that perhaps lag is not the answer. This is problematic.

Novel outbreaks -> This section is intriguing, but the invocation of novel viruses implies that the actual system used is highly diverse, and the models no longer tell the reader what is happening. Note that because the team uses multiple models, each time they invoke a 'simulation' they need to tell the readers which system of equations they are using. This makes it very hard to parse the results.

Next, Figure 3 is quite interesting, but it deserves its own section.

L81: it's -> its

L273: host -> hosts

No table of parameters w/units.

Author's Response to Decision Letter for (RSPB-2021-0435.R0)

See Appendix A.

RSPB-2021-1555.R0

Review form: Reviewer 1

Recommendation

Accept as is

Scientific importance: Is the manuscript an original and important contribution to its field?

Good

General interest: Is the paper of sufficient general interest?

Good

Quality of the paper: Is the overall quality of the paper suitable?

Good

Is the length of the paper justified?

Yes

Should the paper be seen by a specialist statistical reviewer?

No

Do you have any concerns about statistical analyses in this paper? If so, please specify them explicitly in your report.

No

It is a condition of publication that authors make their supporting data, code and materials available - either as supplementary material or hosted in an external repository. Please rate, if applicable, the supporting data on the following criteria.

Is it accessible?

Yes

Is it clear?

Yes

Is it adequate?

Yes

Do you have any ethical concerns with this paper?

No

Comments to the Author

The authors have addressed all my comments with thought. I am pleased to see more information provided about the model set-up and particularly the parameter values. I have no further comments.

Review form: Reviewer 2

Recommendation

Accept with minor revision (please list in comments)

Scientific importance: Is the manuscript an original and important contribution to its field?

Excellent

General interest: Is the paper of sufficient general interest?

Good

Quality of the paper: Is the overall quality of the paper suitable?

Excellent

Is the length of the paper justified?

Yes

Should the paper be seen by a specialist statistical reviewer?

No

Do you have any concerns about statistical analyses in this paper? If so, please specify them explicitly in your report.

No

It is a condition of publication that authors make their supporting data, code and materials available - either as supplementary material or hosted in an external repository. Please rate, if applicable, the supporting data on the following criteria.

Is it accessible?

Yes

Is it clear?

Yes

Is it adequate?

Yes

Do you have any ethical concerns with this paper?

No

Comments to the Author

Summary

The authors have addressed the major points of my review, fixed the central technical error, and have updated the paper in substantive ways. My only last comment is for the authors to consider adding a nuanced statement on the implication of the qualitative fits. A finding of a bounds of a factor of 100 (between a few seconds to multiple minutes) for a lag that is the centerpiece of this article raises some questions on biological mechanisms and raises key open questions. I really wish this was addressed in the Discussion as well where Figure S10 does not make an appearance. In my view, there is a big open question here – if mechanisms of a few seconds lag are genuine, then what is the molecular binding event of that speed that could make that large a difference. And, if the factor is minutes, then how badly do current models fit the data? Right now, perhaps I've missed it, but I don't see on Figure 3 or S10 what happens when one uses the wrong lag? Do the curves stay largely the same or do they diverge? If this paper is to make a difference to the field, rather than potentially declare premature success in totally solving this problem, I think the authors would do their paper and the field a service by being more precise in this specific issue.

Minor points

- Units of adsorption are wrong, they should just be ml/hr. If you want to include clarifying remarks like viruses & cells then take a look at the last line of Equation 1. It is evident that the product of delta and cell density must equal an inverse time, so delta must have units of ml/(cells*hrs) and not involve inverse viruses (see table 1)
- I want to insist that the discussion of lag times includes time units at first introduction and in captions, so that biologists understand the this model continues to insist that somehow lag times of a few seconds – perhaps they are right (certainly the model suggests this answer), but perhaps they are wrong (because perhaps the model is too simplified or is missing key biology), and letting biologically informed readers use their intuition will be possible if the text is clearer that $>10^3$ / hr means that the lag times are on the order of a few seconds. Figure 2 is very helpful in this respect and is far clearer than before. Indeed it makes the point that for lags on the order of minutes then one does not need enormously high viral pressure for CRISPR/SM to coexist. It is only in the case when the lags become *extremely* short that it would seem to favor CRISPR only. The revised presentation makes this far clearer.
- Again, Figure 3 is a fit with a lag of a few seconds, and this is claimed to match the data. Perhaps, but if so, I still wonder if this is right for the right reasons. I still don't understand why the authors have not reported a model with different lag times. Here is the issue, the phage fitness involves a few points, perhaps this model explains it, but presumably many other models could fit these few points. So, it seems important even as a 'Minor' point to add other lags, extending the idea of Figure 2 so that the reader can see whether the present model only fits the data for very short lags (order of seconds) or for longer periods. The caption mentions in passing 'much longer' lag periods, so there is a difference of two orders of magnitude here, which starts to raise questions on what we have learned from these model-data fits.
- Figure 4 d/e have curves that go outside the bounds of the figure. Perhaps this was a compilation/pdf issue but curves should not go outside the bounds. Also Figure 5 figure quality is different than Figure 4 (I prefer the style of 4, with bounding boxes).
- The SI repeatedly uses this phrase “. When all eigenvalues have negative real parts, the equilibrium is stable. If at least one eigenvalue has a real part greater than zero, the equilibrium is unstable (for eigenvalues with real parts equal to zero no determination can immediately be made - though this situation did not come up in our analyses. For further information find standard procedures for linear stability analysis in any text on ordinary differential equations)”. Please just state a simplified version of this earlier and then note that you use this standard approach in all cases and then you should provide a textbook as a citation that you like, e.g., Strogatz or similar
- The references in the SI use a different style than that in the main text; please conform these to the main text style.

Decision letter (RSPB-2021-1555.R0)

13-Sep-2021

Dear Dr Weissman:

Your manuscript has now been peer reviewed and the reviews have been assessed by an Associate Editor. The reviewers' comments (not including confidential comments to the Editor) and the comments from the Associate Editor are included at the end of this email for your reference. As you will see, the reviewers and the Associate Editor are positive but one reviewer has raised some issues that we would like you to address.

Research ethics:

Use of animals and field studies:

It is a condition of publication that you make available the data and research materials supporting the results in the article (<https://royalsociety.org/journals/authors/author-guidelines/#data>). Datasets should be deposited in an appropriate publicly available repository and details of the associated accession number, link or DOI to the datasets must be included in the Data Accessibility section of the article (<https://royalsociety.org/journals/ethics->

policies/data-sharing-mining/). Reference(s) to datasets should also be included in the reference list of the article with DOIs (where available).

Please submit a copy of your revised paper within three weeks. If we do not hear from you within this time your manuscript will be rejected. If you are unable to meet this deadline please let us know as soon as possible, as we may be able to grant a short extension.

Best wishes,
Professor Hans Heesterbeek
mailto: proceedingsb@royalsociety.org

Associate Editor

Comments to Author:

Thank you for extensively revising and resubmitting your paper. It has been assessed by two of the original reviewers. Both reviewers are positive about the revised version and commend you on the comprehensive revisions you have made. There are several outstanding issues for you to deal with that are detailed in the report of Referee #2.

Reviewer(s)' Comments to Author:

Referee: 1

Comments to the Author(s).

The authors have addressed all my comments with thought. I am pleased to see more information provided about the model set-up and particularly the parameter values. I have no further comments.

Referee: 2

Comments to the Author(s).

Summary

The authors have addressed the major points of my review, fixed the central technical error, and have updated the paper in substantive ways. My only last comment is for the authors to consider adding a nuanced statement on the implication of the qualitative fits. A finding of a bounds of a factor of 100 (between a few seconds to multiple minutes) for a lag that is the centerpiece of this article raises some questions on biological mechanisms and raises key open questions. I really wish this was addressed in the Discussion as well where Figure S10 does not make an appearance. In my view, there is a big open question here – if mechanisms of a few seconds lag are genuine, then what is the molecular binding event of that speed that could make that large a difference. And, if the factor is minutes, then how badly do current models fit the data? Right now, perhaps I've missed it, but I don't see on Figure 3 or S10 what happens when one uses the wrong lag? Do the curves stay largely the same or do they diverge? If this paper is to make a difference to the field, rather than potentially declare premature success in totally solving this problem, I think the authors would do their paper and the field a service by being more precise in this specific issue.

Minor points

- Units of adsorption are wrong, they should just be ml/hr. If you want to include clarifying remarks like viruses & cells then take a look at the last line of Equation 1. It is evident that the product of delta and cell density must equal an inverse time, so delta must have units of ml/(cells*hrs) and not involve inverse viruses (see table 1)
- I want to insist that the discussion of lag times includes time units at first introduction and in captions, so that biologists understand the this model continues to insist that somehow lag times of a few seconds – perhaps they are right (certainly the model suggests this answer), but perhaps they are wrong (because perhaps the model is too simplified or is missing key biology), and letting biologically informed readers use their intuition will be possible if the text is clearer that $>10^3$ / hr means that the lag times are on the order of a few seconds. Figure 2 is very helpful in this respect and is far clearer than before. Indeed it makes the point that for lags on the order of minutes then one does not need enormously high viral pressure for CRISPR/SM to coexist. It is only in the case when the lags become *extremely* short that it would seem to favor CRISPR only. The revised presentation makes this far clearer.
- Again, Figure 3 is a fit with a lag of a few seconds, and this is claimed to match the data. Perhaps, but if so, I still wonder if this is right for the right reasons. I still don't understand why the authors have not reported a model with different lag times. Here is the issue, the phage fitness involves a few points, perhaps this model explains it, but presumably many other models could fit these few points. So, it seems important even as a 'Minor' point to add other lags, extending the idea of Figure 2 so that the reader can see whether the present model only fits the data for very short lags (order of seconds) or for longer periods. The caption mentions in passing 'much longer' lag periods, so there is a difference of two orders of magnitude here, which starts to raise questions on what we have learned from these model-data fits.
- Figure 4 d/e have curves that go outside the bounds of the figure. Perhaps this was a compilation/pdf issue but curves should not go outside the bounds. Also Figure 5 figure quality is different than Figure 4 (I prefer the style of 4, with bounding boxes).
- The SI repeatedly uses this phrase “. When all eigenvalues have negative real parts, the equilibrium is stable. If at least one eigenvalue has a real part greater than zero, the equilibrium is unstable (for eigenvalues with real parts equal to zero no determination can immediately be made - though this situation did not come up in our analyses. For further information find standard procedures for linear stability analysis in any text on ordinary differential equations)”. Please just state a simplified version of this earlier and then note that you use this standard approach in all cases and then you should provide a textbook as a citation that you like, e.g., Strogatz or similar
- The references in the SI use a different style than that in the main text; please conform these to the main text style.

Author's Response to Decision Letter for (RSPB-2021-1555.R0)

See Appendix B.

Decision letter (RSPB-2021-1555.R1)

23-Sep-2021

Dear Dr Weissman

I am pleased to inform you that your manuscript entitled "Immune Lag Is a Major Cost of Prokaryotic Adaptive Immunity During Viral Outbreaks" has been accepted for publication in Proceedings B.

Data Accessibility section

Open Access

You are invited to opt for Open Access, making your freely available to all as soon as it is ready for publication under a CC BY licence. Our article processing charge for Open Access is £1700. Corresponding authors from member institutions (<http://royalsocietypublishing.org/site/librarians/allmembers.xhtml>) receive a 25% discount to these charges. For more information please visit <http://royalsocietypublishing.org/open-access>.

Paper charges

Sincerely,
Professor Hans Heesterbeek
Editor, Proceedings B
mailto: proceedingsb@royalsociety.org

Associate Editor:
Board Member
Comments to Author:
(There are no comments.)

Appendix A

Dear Dr Weissman:

I am writing to inform you that your manuscript RSPB-2021-0435 entitled "Immune Lag Is a Major Cost of Prokaryotic Adaptive Immunity During Viral Outbreaks" has, in its current form, been rejected for publication in Proceedings B.

This action has been taken on the advice of referees, who have recommended that substantial and fundamental revisions are necessary. With this in mind we would be happy to consider a resubmission, provided the comments of the referees are fully addressed. However please note that this is not a provisional acceptance.

Sincerely,

Professor Hans Heesterbeek
mailto: proceedingsb@royalsociety.org

Associate Editor
Board Member: 1
Comments to Author:

Your interesting manuscript has been assessed by two expert reviewers. Both found the ideas compelling and potentially interesting. However, each raises some criticisms about the model construction and interpretation. Of particular concern is the second major issue raised by reviewer 2 i.e. that the model extension is fundamentally wrong.

We thank the editor for the chance to revise our manuscript. Please see below for line-by-line responses to the reviewer comments on our manuscript alongside descriptions of changes to the text. These include:

- **An expanded consideration of the form the downregulation term in equations 5 and 6 can take, with a comparison of model behavior with different forms for this term in S5 Text and S7 Figure (in response to reviewer #2's concerns about misspecification)**
- **A much expanded supplement, including a new S1 Text that outlines model assumptions and construction, as well as several new extensions to our model and analysis (S5 and S7 Texts, and S7-S9 and S12-S13 Figures; in response to reviewer #1's request for more detail and some additional information about model behavior)**
- **Modified versions of Figures 1 and 2 that consider the effects of variable costs of the SM mutant (as requested by reviewer #2)**
- **The movement and expansion of Table 1 (formerly S1 Table) to the main text (in response to the request by both reviewers to have the table of parameters in the main text)**

Additionally, we have reworked sections throughout the manuscript to improve clarity based on areas each reviewer indicated as needing more detail to understand.

Reviewer(s)' Comments to Author:

Referee: 1

Comments to the Author(s)

The manuscript considers the question of whether a lag in immune response may be partly responsible for experimental evidence that CRISPR-Cas systems are absent from many host organisms. They develop mathematical models to test their hypothesis, guided by experimental data. They find that indeed a lag in immune response can make CRISPR systems uncompetitive against a non-laggy mutant, even when that mutant carries a cost in reduced growth. The effect is particularly strong for slow-growing and small populations.

The manuscript addresses an interesting question, and the model is used to good effect to make their key arguments. The writing is pretty good - I would say that as a modeller I found it ok, but I wonder if the model construction could do with more detail/motivation for a biological audience. But overall a really nice manuscript that got me thinking.

Despite the insistence that the model is 'simple' - and perhaps it is all that is minimally needed - I think it is actually quite complex to unpick. The more detailed models in the SI are particularly so. I don't really have a major criticism at all - I think this is a very nice piece of work. I guess the most important comment I have is that, as with all models, there are lots of assumptions here and I feel that they could be covered better. Doing so in the SI would probably be fine, but just details like why you chose the chemostat set-up and what that might mean for the results, why you chose the particular forms of growth, infection, downregulation, etc.

We thank the reviewer for pushing us to make our paper more accessible to a less-mathematical audience. We have added a supplementary text that details the many choices involved in building such a model as ours, and explicitly identifies several important decision points. See the new S1 Text for details.

Other than that I just have some thoughts and suggestions that occurred to me while reading the manuscript:

P3 L74 (the model) - I have a couple of questions about the model set-up:

* With your chemostat assumption, a constant volume is only preserved if you can be sure that the total 'outgoing' quantities equal the total 'incoming' quantities. It's not immediately clear that this is true. The rates, w , may be the same, but of course if your incoming quantity is $w \cdot (r_0 + v_0)$ and your outgoing quantity is $w \cdot (R + S + C + I + V)$ these may well be different. It's not even clear to me they are all measured with the same units.

We apologize for the confusion here, which we think is a product of us leaving out some crucial information in our table of parameters. The quantities r_0 , v_0 , R , S , C , I , and V are all concentrations (expressed as microgram/mL, cells/mL, or viral particles/mL), whereas the rate w describes the flow of media into/out of the chemostat. This is standard in chemostat models of host virus interactions (see our citations for early chemostat models and the Quantitative Viral Ecology textbook by Weitz). Thus, volume is conserved over time, though the concentration of individual resource, host, or virus populations will not be. We have added a discussion of this point to S1 Text (section 1.1) and we have also included the quantities R , S , C , I , L , C_F , and V to our table of symbols and definitions (now Table 1) to make this clear.

* COuld you move your paragraph about v_0 higher up - I actually wrote a comment asking about this before finding the relevant text a little later.

Done (see lines 86-94).

* Am I right that you assume a proportion, μ , of infection events actually result in immunisation, and the rest full infection? This is presumably similar to immune priming as seen in plants/insects?

This is the correct interpretation of our model, and is in line with current knowledge of how CRISPR immunization works and other modeling efforts describing CRISPR immunity. While it is outside the scope of this article to discuss relationships with eukaryotic defense systems, we direct the reviewer to several new advances showing links between prokaryotic and eukaryotic defenses (<https://doi.org/10.1038/s41586-020-2762-2>, <https://doi.org/10.1101/2021.01.06.425286>). We have modified the description of μ in the main text to clarify this point (lines 79-80):

“Each time a susceptible host is infected, it may undergo immunization with probability μ to become defended (i.e., spacer-possessing) host (C)”

We have also doubly clarified this point in our supplement (S1 Text section 1.4).

P6 L142 - Can you comment more on your choice of downregulation function as this Holling/Hill type function. It's an interesting form with the 'force of infection' term giving the half-rate threshold. You could perhaps include a couple of plots (in the SI if needed).

We have expanded our analysis of the downregulation function after Reviewer #2 pointed out that, as written in the paper, this function was not correct (we needed an additional parameter). We have now included a correction to the nonlinear downregulation term, and also compared the results of this model against ones with either (1) a linear downregulation term, or (2) no downregulation. We find that the linear and nonlinear forms yield almost identical results, and now include the linear version in the main text for simplicity. See equations 5 and 6 and Lines 143-146:

“Finally, here we assume that cells return from the transcriptionally upregulated state (CF) to the baseline state (C) at a constant rate (ζ), though relaxing this assumption has little effect on the qualitative results of the model (S5 Text and S7 Figure).”

Also see S5 Text and S7 Figure for the comparison of downregulation terms.

P7 L145 - At the start of the results - and in the figure caption for figure 1 - please could you define SM again?

Done.

P7 L146 - The cost here is pretty small - 1% drop in growth? I find it interesting how unlikely it is to win the competition. What's your thinking on this? Presumably the loss due to both lysis and lost growth while infected is not that great? Connected to this, you don't give the parameter values used anywhere in the main text, but I think they should be here rather than just in the SI.

We emphasize that figure 2 shows the long term equilibria of the system. Over long timescales, a strain with a 1% growth defect will go extinct in a competitive setting. Essentially, the cost of lag must exceed the cost of the 1% growth defect for the SM strain to succeed, which only happens when viral titers are high enough that at least 1% of the CRISPR-defended population experiences lag. We thank the reviewer for pointing out a potentially confusing line which we have now rewritten (lines 149-152):

“We found the equilibria of our lag model and determined their stability via linear stability analysis in order to characterize the ultimate outcome of competition between a laggy CRISPR-immune strain and a costly surface mutant (SM) strain (see S6 Text for equations describing equilibria and analysis details).”

We have moved the parameter table S1 to the main text to be Table 1.

P7 L156 - As a modeller I'm interested in how the SM equilibrium obviously appears as a

saddle-node equilibrium, giving rise to the bistable region. This will be to do with the Holling/Hill-type growth rates I guess. Perhaps it is not so interesting biologically though!

We thank the reviewer for pushing us a little here. We agree that this bistability is interesting, though it is not a symptom of the way we model growth as suggested. In fact, we confirmed this by building a model with implicit resource dynamics (logistic growth), which exhibited the same region of bistability. We believe the explanation for the bistability in our model is that at sufficiently high densities, the CRISPR strain is able to rapidly clear viruses from the environment (since it is a DNA-degrading intracellular immune system), which can in turn facilitate it's establishment in these cases by removing the source of lag (i.e., viruses). We support this intuition by building a model in which CRISPR does not draw down viral populations (viral population density is taken to be constant and set at v_0), and are able to show that the region of bistability disappears in this case. This has some interesting biological implications, considering that it is essentially a case of niche construction by the CRISPR strain, though further analysis along these lines is beyond the scope of this study (perhaps something for future work). We include these models in the supplement (S7 Text), and refer to them in the main text (lines 157-161):

“At intermediate lag times and viral densities there was a region of bistability in which the initial conditions of the system determined whether or not it would end up in an SM- or CRISPR-only state, with both equilibria being stable. This bistability is a byproduct of CRISPR's ability to clear viruses from the environment and in doing so reduce the impact of lag when cells are at a high density (S7 Text, S8 Figure, and S9 Figure).”

P8 L168 - Just to be clear, by 'in silico' do you just mean you ran your model for these different titres? If so, how were you measuring fitness here? I'm not going to insist on you doing the work, but it would have been interesting to set it up as a stochastic model and see if the variation ends up anything like from the experiments.

Yes, that is the correct interpretation here, and we have removed the term “in silico” in case it would imply otherwise (lines 178-180):

“We matched these competition experiments to model predictions by solving our model numerically (see Methods section “Simulating Competition Experiments”) to more precisely illustrate this point (figure 3, red line).”

We apologize that our calculation for relative fitness was unclear – we now realize we only mentioned it in the experimental methods and not the simulation methods. The relevant calculation is now described in the text (line 356):

“Bacterial colonies were counted, after which relative fitness was calculated using the equation [29]: (fraction CRISPR strain at 24 hr) × (1 – fraction CRISPR strain at 0 hr) / (fraction CRISPR strain at 0 hr) × (1 – fraction CRISPR strain at 24 hr). (7)”

and (lines 365-366):

“Relative fitness was calculated in the same manner as our experiments, using equation 7.”

Given the large population sizes at the start of these competition experiments and simulations ($\sim 10^7$ cells each of CRISPR and SM strains initially), demographic stochasticity should be essentially irrelevant in this system. Much more likely is that experimentally, it is somewhat difficult to ensure an exact cell density, and noise from experimental error can lead to different outcomes for this system. See our S11 Figure, and we now further emphasize this point in the main text (lines 190-195):

“Viruses and host cells were quantified using serial dilutions, introducing the possibility of multiplicative errors and perhaps making cross-experiment variability less surprising. This cross-experiment variability prevented us from obtaining precise lag estimates (we estimate that $10^{-3} \text{ hr}^{-1} \leq 1/\phi \leq 0.1 \text{ hr}^{-1}$). Initial host density in particular can strongly affect model expectations (S11 Figure). On the other hand, given large starting densities of host ($\sim 10^7$ cells/mL), this variability is unlikely to arise from demographic stochasticity.”

P8 L190 - I find 'non-equilibrium' a bit vague, as it could be that the underlying model cycles (given the free-living viruses and non-linear growth rates it would be possible here) and that would also be termed non-equilibrium. Maybe specify you mean during transient growth or early stages or something?

We thank the reviewer for making a very valid point. We now replace “non-equilibrium dynamics” with the phrase “early, transient dynamics” throughout the text.

P8 L200 - I know the full result is in the SI, but could you state in the main text what the value of kappa is? Is a 10% growth loss very high? Do we have a benchmark to judge that?

We have added the kappa value to the main text and augmented Figure 2 and S13 Figure to include even more severe growth costs (20%, 40%) for comparison (lines 221-222):

“This initial dominance of SM even occurs when the cost of an SM strategy is very high, up to a 20% cost with a short lag and well over a 40% cost with a long lag (S13 Figure).”

Given the rate at which microbes reproduce, a 10% growth deficit is quite high, though what counts as “very high” will always be relative. Certainly it’s not a lethal mutant, which would be a 100% growth deficit. Nevertheless, selection should work quite efficiently to remove a strain with a 10% growth cost since in this case $|s| \gg 1/N_e$, except for very small populations. For reference, this 10% growth deficit means that, in the absence of viruses and calculated from a day of simulated competition against a CRISPR strain in batch culture, the SM strain will have a relative fitness of ~ 0.6 . This is comparable to the higher-end of the distribution of costs of E. coli mutations providing resistance against phage T4 (<https://onlinelibrary.wiley.com/doi/epdf/10.1111/j.1558-5646.1988.tb04149.x>).

P10 Figure 4 - When plotting here, while C is the only cell type to have defences, you can define

the CRISPR-type as being C+CF+L couldn't you? I'm guessing this doesn't make much difference - it looks like L decreases fairly rapidly, and presumably as V decreases so does CF, I just wondered what these total numbers looked like?

We have included a supplementary version of this figure with lagged and un-lagged CRISPR-encoding cell populations added together as requested (S12 Fig). These plots have a log scale on their y-axis so that this aggregation really doesn't change much.

Referee: 2

Comments to the Author(s)

Overall:

This is an interesting paper on the relationship between the strength of viral pressure and the relative benefits of CRISPR-Cas immunity in light of 'lags' (i.e., the time it takes before a system can clear an infection). In doing so, the team assumes immunity is 100% effective in the long-run (in the absence of other mortality factors), but that there is a cost: the time to clear can become excessive in the limit of relatively high infection rates and/or long lags. This becomes an effective growth cost which can be compared against the costs of surface mutations. The model formalism is largely solid, but there are a few major issues.

First, in the absence of parameters, any mathematical analysis, the team has missed an opportunity to explain the self-evident tradeoff between viral pressure (v_0) and lag time ($1/\phi$). Doing so would help to explain the simulation results. I believe that the 'mathematical' results may in fact be contained in the SI, but the paper does not make explicit how they estimated the break points between 'phases'.

We thank the reviewer for identifying a potential point of confusion. This information is, in fact, included in the SI, though it seems we have not sufficiently directed our readers to the correct section of the supplementary text in order to find the relevant analyses. In fact, figure 2 represents such mathematical analysis, and is not the outcome of numerical simulations as the reviewer suggests. We describe our procedure for the model analysis in Figure 2 in S6 Text. Briefly, we find the eigenvalues of the Jacobian matrix of our system and use these to assess the stability of possible equilibria at different combinations of parameter values (i.e., we perform linear stability analysis, a standard mathematical approach). We now clarify this point in the Figure 2 legend:

“Shown are exact analytical solutions of model equilibria characterized via linear stability analysis (see S6 Text).”

We also recognize that the way this figure was described in the main text may have implied figure 2 was the result of simulations, and we have rewritten the relevant text to make it clear that these are analytical, not numerical results (lines 148-163):

“We found the equilibria of our lag model and determined their stability via linear stability analysis in order to characterize the ultimate outcome of competition between a laggy CRISPR-immune strain and a costly surface mutant (SM) strain (see S6 Text for equations describing equilibria and analysis details). Over a wide range of parameter

values the model yielded a single stable CRISPR-immune equilibrium with an extinct SM strain (figure 2). Only when there was extremely high flow of viruses into the system (high v_0) did we see an alternative outcome where the the only stable equilibrium was an SM-only state with an extinct CRISPR strain. For short lag times ($\varphi \geq 10^3 \text{ hr}^{-1}$), the “tipping point” from an all-CRISPR to all-SM state occurred as the external viral pool (v_0) exceeded concentrations of 109 PFU/mL (figure 2). At intermediate lag times and viral densities there was a region of bistability in which the initial conditions of the system determined whether or not it would end up in an SM- or CRISPR-only state, with both equilibria being stable. This bistability is a byproduct of CRISPR’s ability to clear viruses from the environment and in doing so reduce the impact of lag when cells are at a high density (S7 Text, S8 Figure, and S9 Figure). In no case did the two strains, CRISPR and SM, coexist stably over the parameter regimes considered.”

We hope the reviewer understands the difficult balance that theory papers are expected to accomplish – with enough math in the main text to satisfy the more mathematically inclined reader, but not so much as to discourage the wet-lab biologist.

Second, the extension of the model to include upregulation is mis-specified, i.e., it is wrong. The denominator of the ‘downregulation’ term adds up a rate with a density. That is not possible. Now perhaps there is some typo or reason for this error, but in my view, this model extension and results are incorrectly specified. I have cross-checked Table 1 (which itself misses multiple units and mis-specifies the units of the adsorption rate).

We thank the reviewer for their attention to detail. As written, this equation was indeed incorrect and we have modified it accordingly (via the addition of a missing parameter). In fact, this comment inspired us to look more deeply at the downregulation term, and we found very similar dynamics using a much simpler linear term, which we now implement in the main model in place of the non-linear downregulation term. We also include a comparison of several forms for the downregulation term (no downregulation, linear downregulation, and non-linear downregulation) in the supplement and show that the shape of this term is not particularly relevant to our main qualitative result (i.e., that upregulation can rescue the population from lag). See lines 143-146:

“Finally, here we assume that cells return from the transcriptionally upregulated state (CF) to the baseline state (C) at a constant rate (ζ), though relaxing this assumption has little effect on the qualitative results of the model (S5 Text and S7 Figure).”

and supplement (S5 Text and S7 Figure).

The adsorbtion rate units have been fixed and Table 1 (previously S1 Table) cleaned up. We apologize, apparently this table had not been updated to reflect the most current version of the model and is now correct and expanded.

Finally, the key (and brief) result is simply that when there is lag, then high viral pressure means that CRISPR-Cas strains are worse off than surface mutation strains – but that this ultimately depends on inherent costs of growth, lag time, and viral intensity. Hence, the exciting direct piece is a link between theory and figure 3 (a new experiment). Yet, for all the talk of lag, the

data seems to be explained with basically 0 lag (if I assume that the units are hrs, then 10^3 for ϕ implies 1/1000 of an hour lag, or essential 0 lag). If that is true, then how important is lag really?

We agree with the reviewer that it is surprising that such a short lag could so severely impact population dynamics, and yet our models show unequivocally that it can (e.g., Fig 4, compare panels a and b). In fact, we think this is what makes our findings so interesting. Key here is understanding that lag only has an effect when viral titers are very high, such that all or nearly all of the host population faces viral challenge, and possibly does so multiple subsequent times. These multiple rounds of infection that occur when viruses greatly outnumber hosts can cause a severe lag. Is this likely to happen in real life? Yes, it should happen early on during an outbreak of novel viruses, when the population of CRISPR-defended host cells is relatively small (hence the motivation for much of the latter part of the manuscript, e.g., Figure 4). We now clarify this point in the main text (lines 198-205):

“Interestingly, the lags estimated in the last section are quite short, ranging from 3.6 seconds ($\phi = 10^3 \text{ hr}^{-1}$) to 6 minutes ($\phi = 10 \text{ hr}^{-1}$). This short length in part explains why such high viral titres (often greater than 109 PFU/mL) are required to observe any effect of lag on the host population. The cost of lag seems to only revealed when immune hosts are facing multiple subsequent viral infections. When will such high viral titres be achieved? We suspected that during a viral outbreak where only a small fraction of the host population is initially immune, viral titres might greatly exceed immune host densities and lead to a clear cost of lag.”

It could also be that CRISPR immunity is not 100% perfect, and at these high levels, then such imperfect defense could also lead to a fitness difference.

Previous work has shown that this is not the case, as we discuss and now clarify in the manuscript (lines 26-31):

“Importantly, while CRISPR-immune cells were observed to have reduced fitness when exposed to phage in competition experiments, subsequent efficiency of plating experiments showed that CRISPR-immune cells did not experience a detectable level of phage-induced mortality [29], indicating that phage inhibit the growth of immune cells but either do not kill them or do so very rarely at levels that cannot explain CRISPR’s inducible cost.”

Also see (lines 183-185):

“Importantly, the original work by Westra et al. [29], showed that inducible cost of CRISPR-Cas immunity was not due to virus-induced mortality, as even less-fit CRISPR-immune cells survived at high viral titres.”

There are multiple leaps of faith to explain data that doesn’t fit the model and in fact the bulk of the SI includes an analysis of a different more complex model.

It is true the supplement contains a large number of analyses of model variants (even more variants in this round of revision). We include these with the explicit goal of showing that our results are robust to different assumptions one can make about CRISPR immunity and virus-host interactions. We view this not as a sweeping of information under the rug, but rather as us doing our due diligence to show that our results are not oddities due to any particular set of modeling assumptions.

In sum: the core idea is compelling, but the gap between model and experiment and mis-specification of a core component detracts (and indeed, the error in part of the model renders part of the results unreviewable at this stage).

Major Revision:

Model framework – Part 1 – Presentation

The overall structure is sensible and well described. But, improvements would be welcome. Figure 1 is nice, but also misses a key point -> the advantage of surface mutations are evident when there is no cost to such mutations, otherwise, even rapid reproduction may be slower than the lag. Also, title should be 'Lagged immunity' and not 'Laggy Immunity'. Hence, note that the growth cost parameter (κ) in equation 2 would imply alternative cases for the right hand side of Figure 1, one in which many more cells were made and one in which very few were made. Note that the first part of the results assumes $\kappa = 0.01$, implying that it is critical to include two versions of the κ parameter in the schematic. A schematic should also include a standard compartmental model view of the dynamics, especially including the addition of Eqs. 3 and 4 that imply different variants of the framework.

We thank the reviewer for their advice on figure construction – the new and improved Figure 1 now includes different costs as well as a wiring diagram for the lag model. We also include a diagram for the base model without lag an expanded diagram for the version of the model with transcriptional upregulation in the supplement (S1 and S6 Figures).

Model framework – Part 2 – Misspecification of the upregulation component

Indeed, I am confused by the meaning of equations (5) and (6). This variant of the model seems to retain a lagged state, which then goes into a fast immunity state and then a slow immunity state. But, strangely enough, in Equation (6) the immune cells seem just as susceptible as susceptible cells (w/out CRISPR-Cas immunity).

Indeed the “infection” rate is the same for immune (C) and susceptible (S) cells, but the fates of these cells are quite different. The C cells will be converted at a rate $\delta \cdot V \cdot C$ to the lagged cell state, whereas the S cells will be converted at a rate $\delta \cdot V \cdot S$ to the infected cell state. Note that the term in question is also present in the original lag model without upregulation (eqn 4). To clarify this point, we have changed the label on this term in the equation to be “enter lag” instead of “infection”. Also see the wiring diagram in Figure 1 (as requested above) and S1 and S6 Figures, which should help readers track the fate of cells through the system.

In addition, Equations (5) and (6) are mis-specified and have a technical error. Note that in the effort to downregulate the downregulation rate, the team uses a denominator that adds up two variables with different units. Cf has units of density, whereas ΔV has units of inverse time. These two values can't be added in this way – it's simply a mis-specified model equation. If the team used this particular setting, then any/all results using this model equation must be revised. If this is a typo, then the team should fix it, but my feeling here is that the model is in fact mis-specified. As stated, all model results with upregulation of the CRISPR locus have a mis-specified model (e.g., later Results and Figure 5).

See our earlier comment addressing the misspecification of the upregulation component. This term has now been changed to a linear term and we include a supplemental discussion and comparison of different forms for the downregulation term (see S5 Text and S7 Figure). We note that changing the form of the downregulation term to a linear form had essentially no effect on our overall results (the point we wanted make that upregulation of the CRISPR locus could potentially mitigate lag is a simple one).

Result description

SM vs. Lagged CRISPR

The presentation of Figure 2 w/out dynamics precludes understanding.

We are not sure we understand this comment - it is quite standard to analyze model equilibria in order to characterize model behavior. We wonder whether the way this section had previously been written implied that Figure 2 was the result of simulations, and we have rewritten the discussion of this figure in the main text to make clear that this is not the case (see earlier comments about Figure 2). In any case, we agree with the reviewer that dynamics are an important part of model analysis, and in the case of our system the transient dynamics are extremely important. We dedicate much of the later half of the paper to understanding these transient dynamics.

In addition, a choice of a virostate model is unusual.

Our virostat model is simply a generalization of a typical chemostat model (set $v_0=0$ for the classical model) that allows for viral immigration. See our previous discussion of this choice in the main text (lines 87-94):

“Observe that in a small departure from the classical chemostat model we allow constant immigration of viruses into the system from some environmental pool (v_0). This is an entirely experimentally tractable modification (e.g., by adding set concentrations of virus to the resource reservoir), and better represents natural systems which are not closed and where hosts likely face constant challenges in the form of newly-arriving viruses. Note that this basic model only considers a single viral genotype, so that immune hosts will also be immune to immigrating viruses (though see outbreak simulations discussed later for simulations in which this is not the case). For traditional continuous culture without viral inflow simply let $v_0 = 0$.”

and also now in the supplement (S1 Text, section 1.1)

There is also no table of parameters.

This is not the case, though we recognize this supplementary table was not adequately referenced in the main text in the previous version of the manuscript. We have moved this table to the main text (now Table 1) at the request of both reviewers.

And, I fully expect that the transition line will change w/changes in kappa. Note that the effective growth cost here is not analyzed. That is a missed opportunity.

A related analysis had been done in Figure S2 with a more complex model, as well as in S13 Figures for our analysis of transient model dynamics for the model in the main text, but we now also include this analysis in Figure 2 at the reviewer's request (see added transition lines). The transition line does change with an increase in cost, though not by as much as one might imagine nor in a particularly interesting way (see new Fig 2). Higher cost simply shifts the transition line in the y-direction (higher v_0 required to cause CRISPR to be disfavored at equilibrium). The overall implications stay the same.

SM hosts have a growth cost of kappa. Whereas, when $V=v_0$, then there is an intrinsic switch to lagged states (ΔV_0) and then an intrinsic rate of return (ϕ). Why did the authors not try and combine these to construct a simplified theory for the transition (which clearly must have lag length be small when V_0 is high to avoid the continual push into the lagged state leading to a severely depressed growth). The authors say 'analyzed our model', but I don't know if that's mathematical analysis or computation. I don't see any math, so I can only presume they did not in fact come up with the theory that seems evident from the nature of the transitions.

This requested theory is, in fact, visualized in Figure 2, which directly references (in the figure legend) the supplementary text in which the mathematics used to analyze our model can be found (S6 Text). See our earlier comments about Figure 2 which largely address this concern, and help connect the analyses in the supplement to the main text via edits to the Figure 2 legend as well as the first paragraph of the results. Briefly, we find equations describing the equilibria of our model and then perform linear stability analysis (by calculating the eigenvalues of the Jacobian) to assess the stability of those equilibria at different parameter combinations. The result is Figure 2, which is the output of this analysis with different combinations of v_0 and ϕ . We have made changes to the Figure 2 legend and description in the main text (lines 148-163) to help clarify this point (see earlier comments about Figure 2).

Importantly, V does not equal v_0 at all equilibria as the reviewer suggests, since CRISPR clears virus and will suppress the viral population somewhat (although $V=v_0$ in the SM-only regime).

The transition lines seem jagged at times, so I am not sure what this figure actually represents.

This was a side effect of the visualization approach we used, which was done over a relatively coarse grid. The lines in the updated version are smooth. Again, we hope that

our revised discussion of this figure in the main text has clarified the meaning of this figure (lines 148-163).

There are also no units on Figure 2 (unacceptable, throughout the use of units must be included if the team is going to compare across data/models, what is 10^3 for phi exactly in terms of a rate). Now, having reviewed Table S1 my understanding is that the authors use a short lag time of $\phi = 10^3$. That would seem to correspond to an average lag time of 1/1000 hrs or approximately 3 seconds. What does that mean biologically? For a model that is trying to explain an outcome w/lag to find that the lag is ~ 3 seconds implies that perhaps lag is not the answer. This is problematic.

Figure 2 now has units labeled on both axes. See our response to earlier comment addressing the point about the length of lag and making clarifying changes to the main text (lines 198-205). We emphasize that our model shows that even an extremely short lag can be quite costly when viral titres are high, and that this situation occurs during viral outbreaks. Thus, we show that indeed a short lag is the answer, due to the threat of many back-to-back infections. We also have added units where needed throughout the text in this version of the manuscript.

Novel outbreaks -> This section is intriguing, but the invocation of novel viruses implies that the actual system used is highly diverse, and the models no longer tell the reader what is happening. Note that because the team uses multiple models, each time they invoke a 'simulation' they need to tell the readers which system of equations they are using. This makes it very hard to parse the results.

Even in the least diverse of natural systems, it is likely that a new virus will immigrate by chance at some point. We eschew discussions of diversity in the current manuscript, but we show that even in cases where CRISPR-immune strains should "win", with a perfect idealized CRISPR system, during an outbreak CRISPR may not be the optimal means of defense because of lag. We apologize for any lack of clarity - we have now modified the text in this section and more explicitly referenced the relevant section of the Methods that describes each simulation scenario throughout the text (lines 206-211):

"We simulated an outbreak of "novel" virus to which preexisting CRISPR-Cas immunity did not exist in the population, or to which only a very small proportion of the population was already immunized (see Methods section "Simulating Outbreaks"). In practice, this was achieved by initializing the system with a dense susceptible population (10^8 cells/mL) and a very small CRISPR immune population (100 cells/mL), both exposed to a small environmental viral pool ($v_0 = 100$ viruses/mL), and solving numerically."

Next, Figure 3 is quite interesting, but it deserves its own section.

With respect, we disagree that splitting up this section of the text would aid in readability, and as such have left the section as-is. We hope the reviewer understands that we are in a position where the Proceedings B article length constraints limit the amount we can add to the manuscript at this time that isn't absolutely essential.

L81: it's -> its

L273: host -> hosts

Fixed.

No table of parameters w/units.

See previous comment (the table of parameters was previously S1 Table, now cleaned up and moved to be Table 1).

Appendix B

Associate Editor

Comments to Author:

Thank you for extensively revising and resubmitting your paper. It has been assessed by two of the original reviewers. Both reviewers are positive about the revised version and commend you on the comprehensive revisions you have made. There are several outstanding issues for you to deal with that are detailed in the report of Referee #2.

We thank the editor for the chance to revise our manuscript. Please see our point-by-point responses below, including:

- **A modified Figure 3 that shows multiple lag lengths for the model**
- **A new S11 figure that directly compares the data from the three experiments discussed and different lag length assumptions in the model**
- **Additions to the discussion that discuss parameter uncertainty and potential avenues of future research**

Reviewer(s)' Comments to Author:

Referee: 1

Comments to the Author(s).

The authors have addressed all my comments with thought. I am pleased to see more information provided about the model set-up and particularly the parameter values. I have no further comments.

Referee: 2

Comments to the Author(s).

Summary

The authors have addressed the major points of my review, fixed the central technical error, and have updated the paper in substantive ways. My only last comment is for the authors to consider adding a nuanced statement on the implication of the qualitative fits. A finding of a bounds of a factor of 100 (between a few seconds to multiple minutes) for a lag that is the centerpiece of this article raises some questions on biological mechanisms and raises key open questions. I really wish this was addressed in the Discussion as well where Figure S10 does not make an appearance. In my view, there is a big open question here – if mechanisms of a few seconds lag are genuine, then what is the molecular binding event of that speed that could make that large a difference. And, if the factor is minutes, then how badly do current models fit the data? Right now, perhaps I've missed it, but I don't see on Figure 3 or S10 what happens when one uses the wrong lag? Do the curves stay largely the same or do they diverge? If this paper is to make a difference to the field, rather than potentially declare premature success in totally solving this problem, I think the authors would do their paper and the field a service by being more precise in this specific issue.

We agree with the reviewer that it is important to understand the range of possible lags when considering the implications of our models. It is for this very reason we considered both long and

short lags in detail in figure 4, in order to demonstrate that the population level consequences of lag can be severe for both short and long lags. We agree, though, that it would be good to also show a direct comparison of the experimental data in Figures 3 and S10. We have added a new supplementary figure (S11 Fig) that makes this comparison and overlays the range of lags considered in the manuscript to help give the reader a sense of this variability (lines 186-189):

“Finally, we note that while the qualitative results of these competition experiments are highly reproducible, with a steep decrease in the fitness of CRISPR-Cas immune strains occurring at high MOI, where exactly this transition occurs and the baseline relative fitness of the CRISPR-immune strain in the absence of virus appear to be quite variable between replicates and experiments (figure 3, S10 Figure, and see S11 Figure for a direct comparison between the two).”

We have also modified figure 3 to show how our model behaves under a range of lag times. We have also added a more direct discussion of this variability and its importance for future investigations to the discussion as requested (lines 328-336):

“Similar to the uncertainty surrounding the mechanisms causing lag, we do not have a good estimate for the length of the lag period. Our estimated lag durations from different experiments ranged over two orders of magnitude (from seconds to minutes; figure 3, S10-S11 Figures), even though the same strains were used across experiments [30, 29]. The competition experiments we used to parameterize our model likely lack the precision for an accurate estimation of lag duration (S12 Figure). Alternative experimental approaches that more directly assess growth slowdowns (e.g., single cell analyses using microfluidic devices [74]) may be required to obtain accurate parameter estimates. We analyzed the population-level implications of immune lag, but much is left to be done in order to characterize the cellular-level mechanisms and effects of lag.”

Minor points

- Units of adsorption are wrong, they should just be ml/hr. If you want to include clarifying remarks like viruses & cells then take a look at the last line of Equation 1. It is evident that the product of delta and cell density must equal an inverse time, so delta must have units of ml/(cells*hrs) and not involve inverse viruses (see table 1)

Units have been changed as requested, apologies for the confusion.

- I want to insist that the discussion of lag times includes time units at first introduction and in captions, so that biologists understand the this model continues to insist that somehow lag times of a few seconds – perhaps they are right (certainly the model suggests this answer), but perhaps they are wrong (because perhaps the model is too simplified or is missing key biology), and letting biologically informed readers use their intuition will be possible if the text is clearer that $>10^3$ / hr means that the lag times are on the order of a few seconds. Figure 2 is very helpful in this respect and is far clearer than before. Indeed it makes the point that for lags on the order of minutes then one does not need enormously high viral pressure for CRISPR/SM to coexist. It is only in the case when the lags become *extremely* short that it would seem to favor CRISPR only. The revised presentation makes this far clearer.

We have now added lag length estimates in terms of time to all places where we discuss the parameter ϕ as requested. As discussed at length in our last response, we would like to emphasize that the population-level consequences depicted in figure 4 are true even when lags are very short because phage titers are so high during an outbreak. A few seconds can make a big difference when the chance of infection and reinfection is very high.

- Again, Figure 3 is a fit with a lag of a few seconds, and this is claimed to match the data. Perhaps, but if so, I still wonder if this is right for the right reasons. I still don't understand why the authors have not reported a model with different lag times. Here is the issue, the phage fitness involves a few points, perhaps this model explains it, but presumably many other models could fit these few points. So, it seems important even as a 'Minor' point to add other lags, extending the idea of Figure 2 so that the reader can see whether the present model only fits the data for very short lags (order of seconds) or for longer periods. The caption mentions in passing 'much longer' lag periods, so there is a difference of two orders of magnitude here, which starts to raise questions on what we have learned from these model-data fits.

See updated figure three which now shows how lag different lag times in the model do or do not match up with the data, as requested. We agree that it would be great to have precise lag estimates but we do not think this is likely to be possible using the current methods (see our added bit of discussion in response to the reviewer's main point above which mentions this).

- Figure 4 d/e have curves that go outside the bounds of the figure. Perhaps this was a compilation/pdf issue but curves should not go outside the bounds. Also Figure 5 figure quality is different than Figure 4 (I prefer the style of 4, with bounding boxes).

Apologies, this is a persistent issue with R's pdf output and only appears when viewing the pdf within some programs (so we did not see it on our computers). The updated version should not have this problem. We have also changes the style of Figure 5 as requested.

- The SI repeatedly uses this phrase “. When all eigenvalues have negative real parts, the equilibrium is stable. If at least one eigenvalue has a real part greater than zero, the equilibrium is unstable (for eigenvalues with real parts equal to zero no determination can immediately be made - though this situation did not come up in our analyses. For further information find standard procedures for linear stability analysis in any text on ordinary differential equations)”. Please just state a simplified version of this earlier and then note that you use this standard approach in all cases and then you should provide a textbook as a citation that you like, e.g., Strogatz or similar

We have modified the SI to limit our use of this phrase to a single instance and added a reference as requested.

- The references in the SI use a different style than that in the main text; please conform these to the main text style.

Done.